# Reinforcing LLM Agents via Policy Optimization with Action Decomposition

**Muning Wen[1], Ziyu Wan[1], Jun Wang[2], Weinan Zhang[1,†], Ying Wen[1,†],**
[1]Shanghai Jiao Tong University, [2]University College London

## Abstract

Language models as intelligent agents push the boundaries of sequential decision-making agents but struggle with limited knowledge of environmental dynamics and exponentially huge action space. Recent efforts like GLAM and TWOSOME manually constrain the action space to a restricted subset and employ reinforcement learning to align agents' knowledge with specific environments. However, they overlook fine-grained credit assignments for intra-action tokens, which is essential for efficient language agent optimization, and rely on human's prior knowledge to restrict action space. This paper proposes decomposing language agent optimization from the action level to the token level, offering finer supervision for each intra-action token and manageable optimization complexity in environments with unrestricted action spaces. Beginning with the simplification of flattening all actions, we theoretically explore the discrepancies between action-level optimization and this naive token-level optimization. We then derive the Bellman backup with Action Decomposition (BAD) to integrate credit assignments for both intra-action and inter-action tokens, effectively eliminating the discrepancies. Implementing BAD within the PPO algorithm, we introduce Policy Optimization with Action Decomposition (POAD). POAD benefits from a finer-grained credit assignment process and lower optimization complexity, leading to enhanced learning efficiency and generalization abilities in aligning language agents with interactive environments. We validate POAD across diverse testbeds, with results affirming the advantages of our approach and the correctness of our theoretical analysis[1].

## 1 Introduction

Large language models (LLMs) have demonstrated promising capabilities of solving various tasks, from instructions following to complex reasoning and real-world interaction [1–3]. This growing task-solving ability underscores their potential as intelligent language agents in interactive environments [4, 5]. However, despite in-context language generation aiding comprehension of environmental states and action spaces in sequential decision-making tasks, misalignment issues [4, 5] such as generating invalid actions and lacking knowledge of environmental dynamics hinder these agents' ability to complete decision-making tasks robustly and efficiently.

Recent advances [4–8] have showcased that the aforementioned challenges can be alleviated in a trial-and-error learning style, namely Reinforcement Learning (RL) [9]. Representatively, GLAM [4] and TWOSOME [5] treat the language actions, i.e. token sequences outputted by a language model, as whole units, and optimize actions' likelihood, calculated as the products of conditional probabilities of intra-action tokens. Leveraging the chain rule of probability, they build the bridge between optimizing actions and optimizing tokens, aligning the training process of language models as

---

[1][†]Corresponding to Ying Wen <ying.wen@sjtu.edu.cn>, Weinan Zhang <wnzhang@sjtu.edu.cn>. The source code could be accessed directly with this link `https://github.com/morning9393/ADRL`.

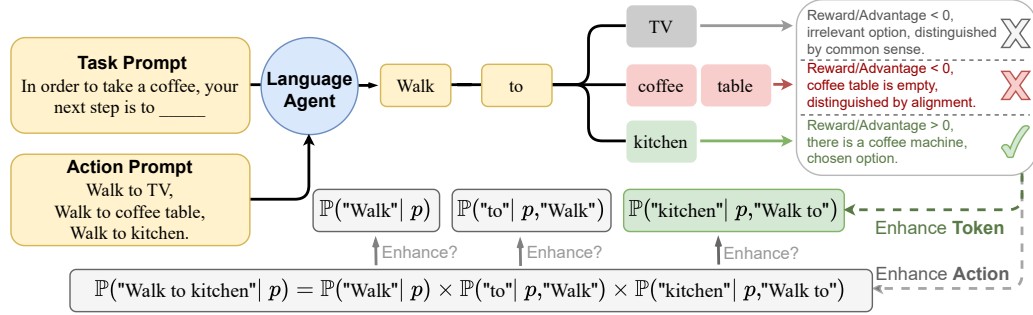

Figure 1: A Case to demonstrate: (a) the necessity of aligning language agents with environments to exclude the wrong option, since the agent does not initially know that *"coffee table is empty"*. (b) Action-level optimization is uncertain to what extent the key tokens, i.e. $\mathbb{P}$("kitchen"$|p$, "Walk to"), will be enhanced when optimizing the joint probability $\mathbb{P}$("Walk to kitchen"$|p$).

next-token predictors with the RL objective of maximizing actions' utilities. However, they still suffer from limitations in optimization and exploration efficiency, due to the uncertainty of credit assignment to intra-action tokens. As shown in Figure 1, when optimizing the distribution over three candidate actions that only differ from the last token, action-level policy optimization strategies in previous works cannot ensure that the probability of the key tokens, i.e. $\mathbb{P}$("kitchen"$|p$, "Walk to") here, will be enhanced precisely when optimizing the joint probability $\mathbb{P}$("Walk to kitchen"$|p$). Furthermore, optimizing at the action level poses the challenge of overlarge optimization complexity due to exponentially growing action spaces, leading GLAM and TWOSOME to manually constrain action spaces. But they remain incapable in environments whose action spaces cannot be restricted.

A natural attempt to solve these problems is to incorporate the token generation process in each decision step as part of the sequential decision-making process [10–12]. This approach resolves the credit assignment problem and reduces the growth of optimization complexity from multiplicative to additive as the number of intra-action tokens increases, by updating each token's output distribution separately with fine-grained signals from value backups. Empirical success in many single-step tasks, e.g. question answering [13] and alignment [10, 11], have demonstrated its effectiveness. However, the multi-step nature of sequential decision-making tasks leads to extra difficulties in coordinating the credit assignment process across actions and their constituent tokens. In such scenarios, embedding the intra-action token generation process into the original Markov Decision Process (MDP) [14] results in a new MDP inconsistent with the original one. Intuitively speaking, it introduces an undesired assumption of "*in a sentence, tokens appearing later are more important for conveying meaning than those appearing earlier.*" This assumption, however, is unrealistic due to the nature of linguistic actions. Such a significant discrepancy has been ignored in previous research and has not been tackled properly for a long time.

In this paper, we provide a comprehensive theoretical analysis of the discrepancy and derive the Bellman backup with Action Decomposition (BAD), which guarantees theoretical consistency with optimizing the original MDP. While being possible to seamlessly integrate with a variety of traditional RL methods, in this work, we apply BAD to Proximal Policy Optimization (PPO) [15], resulting in the formulation of Policy Optimization with Action Decomposition (POAD). Benefiting from the finer-grained supervision afforded by BAD, POAD mitigates the uncertainty in the credit assignment process described in Figure 1, thereby enjoying better interpretability, lower optimization complexity, and higher training efficiency. Meanwhile, it theoretically maintains consistency between the token-level training process for language models and the RL objective of maximizing actions' utilities. We justify our claims by evaluating POAD in both classical sequential decision-making environments with limited action space, i.e Overcooked and VirtualHome [5], and a self-constructed data science coding environment featuring an unrestricted action space, i.e. DataSciCoding; results verify POAD's advantages in performance and efficiency over baseline methods, highlighting the significance of BAD. Moreover, we empirically demonstrate that language agents trained with POAD exhibit excellent generalization ability across unseen tasks, without compromising the inherent functionalities of language models. Finally, ablation studies confirm the correctness of our theoretical insights.

## 2 Related Works

**Language-based Decision-Making Agents.** Leveraging LLMs' powerful capability and plenty of common knowledge, recent efforts successfully adapt LLMs in decision-making tasks as a policy model in interactive environments. In robotics, LLMs have been employed as high-level planners of control policies [16–18]. Similarly, LLMs work particularly well in text-based environments [19, 20]. ReAct [21] combines chain-of-thought reasoning [2] with acting-based prompt, efficiently solving hard decision-making tasks. Self-Refine [22] and Reflexion [23] further improve language agents' efficiency and robustness via online adaptation through the self-reflection process. In this work, we also apply language agents in sequential decision-making scenarios, i.e. interactive environments.

**Fine-tuning Language Models with RL.** A body of literature explores the prospects of leveraging strategic planning methodologies to enhance the performance of language agents [7, 24, 25, 6]. Besides, RL has also been widely applied in fine-tuning LLMs [10, 11, 7, 24, 25, 6]. Particularly, proximal policy optimization (PPO) [15] is the most commonly used RL method for reinforcement learning from human feedback (RLHF), proposing a breakthrough in aligning LLMs with human preference [10, 11]. In classical sequential decision-making scenarios, to align the objective of token-level optimization with action-level optimization, GLAM [4] and TWOSOME [5] estimate the probability of possible actions with the products of the conditional probability of tokens composing the actions and update the action as a whole. In this work, instead of treating an action as a whole, we attempt to decompose actions and explicitly assign precise credit to each intra-action token, while ensuring that its optimality is consistent with updates at the action level. While a concurrent work, ArCHer [26], also targets token-level supervision for LLMs in interactive environments, it employs a hierarchical RL framework, using a Q-network for action-level credit approximation and REINFORCE [27] for token-level backpropagation. However, ArCHer's use of multiple value networks (Q, V, and optional baseline networks) demands extensive manual tuning and can introduce cumulative bias and variance, potentially affecting stability.

**Sequential Decomposition in RL.** Recent years have witnessed an increasing trend of decomposing high-dimension actions to leverage the powerful modeling abilities of sequential models like Transformer [28] in RL problems [29–34]. While Kuba et al. [35, 36] proposing a sequential decomposition method to provide finer-grained supervision for multi-agent joint actions, Multi-agent Transformer [31] inherits this idea and solves multi-agent RL problems with Transformer. More recently, Q-transformer [34] managed to decompose the Q-functions for high-dimensional actions by representing each action dimension as separate tokens. For language agents, the language generation process inherently conforms to the pattern of sequential decomposition, which offers a promising avenue for providing finer-grained supervision to intra-action tokens.

## 3 Preliminaries

### 3.1 Language-augmented RL

In this work, we assume a textual RL setting that frames sequential decision-making problems with linguistic inputs and outputs as a language-augmented Markov Decision Process (MDP) $\mathcal{M} = (\mathcal{V}, \mathcal{S}, \mathcal{A}, \mathcal{T}, R, \gamma)$ [37, 4]. Given $\mathcal{V}$ the vocabulary and $w \in \mathcal{V}$ the tokens, $\mathcal{A} \subset \mathcal{V}^N$, $\mathcal{S} \subset \mathcal{V}^N$ are the action space and state space respectively, i.e. actions and states are sequences of tokens. $\mathcal{T} : \mathcal{S} \times \mathcal{A} \mapsto \mathcal{S}$ is the state transition function. $R : \mathcal{S} \times \mathcal{A} \mapsto \mathbb{R}$ is the reward function that only responds to complete actions, and $\gamma$ is the discounted factor that typically less than 1. At time step $t \in \mathbb{N}$, a language agent receives a textual state $s_t \in \mathcal{S}$ from an interactive environment as input and generates an action $a_t \in \mathcal{A}$ in an auto-regressive manner, i.e. $a_t = (w_t^1, \ldots, w_t^{|a_t|})$ where $|a_t|$ denotes the number of tokens in the action string and $\{w_t^i\}_{i=1}^{|a_t|}$ are tokens in it. Then, textual action $a_t$ will be grounded to a specific API call or command in the environment [38]. After execution, the language agent receives a reward $r_t = R(s_t, a_t)$ along with the next state $s_{t+1}$, based on the transition function $\mathcal{T}$. Following this process with trajectories of a maximum timestep $T$, the agents earn a discounted cumulative return of $R^\gamma = \sum_{t=0}^T \gamma^t r_t$, which is aimed to be maximized by RL algorithms.

## 3.2 Action-level Policy Optimization

We begin by briefly reviewing the process of action-level policy optimization which is widely adopted in several state-of-the-art methods that align language agents with environments via RL algorithms [4, 5, 39]. It facilitates seamless integration between any textual environment and conventional RL algorithms and thus is an ideal starting point for our analysis.

The possibly achieved episodic return following policy $\pi$ given action and state is usually evaluated by state-action value function $Q_\pi(s, a)$ or state value function $V_\pi(s)$. Then, a language agent updates its policy $\pi$ according to credits calculated on the value functions, defined as

$$Q_\pi(s, a) \triangleq \mathbb{E}_{s_{1:T} \sim \mathcal{T}, a_{1:T} \sim \pi} \big[ R^\gamma | s_0 = s, a_0 = a \big], \tag{1}$$

$$V_\pi(s) \triangleq \mathbb{E}_{s_{1:T} \sim \mathcal{T}, a_{0:T} \sim \pi} \big[ R^\gamma | s_0 = s \big], \tag{2}$$

where $a = (w^1, \ldots, w^{|a_t|}) = w^{1:|a|}$. While Ahn et al. [16] builds the connection between the likelihoods of actions and tokens through the chain rule of probability as

$$\pi(a|s) = \prod_{j=1}^{|a|} \pi(w^j | s, w^{1:j-1}), \tag{3}$$

recent approaches like GLAM [4] and TWOSOME [5] leverage similar ideas and optimize action-level likelihoods with RL methods directly. When considering optimizing for $\pi(a|s)$, Equations 1 and 2 are aligned with the definition of the value function in traditional RL settings, allowing them to be updated with traditional Bellman backup [9]

$$Q_\pi(s_t, a_t) \leftarrow R(s_t, a_t) + \gamma \max_{a_{t+1}} Q_\pi(s_{t+1}, a_{t+1}), \tag{4}$$

$$V_\pi(s_t) \leftarrow R(s_t, a_t) + \gamma V_\pi(s_{t+1}). \tag{5}$$

Moreover, it is noteworthy that Equation 3 calculates the likelihood of action $a$ in an exponentially growing language space as $|a|$ increases, i.e. $|\mathcal{A}| = |\mathcal{V}|^{|a|}$. Exploration and optimization in such a huge action space are typically intractable for RL methods. Therefore, in the settings of GLAM and TWOSOME, the feasible action space is significantly restricted and smaller than the entire language space, i.e. $|\mathcal{A}| \ll |\mathcal{V}|^{|a|}$. Taking TWOSOME as an example, it optimizes the likelihood of action $a$ concerning the feasible action space with Equation 6 to mask out invalid outputs.

$$\pi_{\text{TWOSOME}}(a|s) = \frac{\exp(\log \pi(a|s)/L(a))}{\sum_{a' \in \mathcal{A}} \exp(\log \pi(a'|s)/L(a'))}. \tag{6}$$

$L(a)$ indicates the number of tokens or words in the action prompt, utilized as a normalization term to mitigate the effects of varying action lengths.

Underpinned by Equation 3, GLAM and TWOSOME ensure the consistency between token-level optimization for language models and action-level optimization in an RL manner, without the need to explicitly assign credits for intra-action tokens. However, the jointness of the objective causes difficulties associated with the uncertainty in the credit assignment process [40, 41]—as shown in Figure 1, after assigning credit to an action, it's unsure whether key tokens in this action have been identified, and how much they are influenced. Thus, conducting RL training at the action level introduces uncertainty, which may lead to an inefficient learning process for the language agent.

# 4 From Actions to Tokens: Naive Token-level Policy Optimization

## 4.1 Naive Token-level Policy Optimization

To address the unclear credit assignment issue described in Figure 1 and Section 3.2, our target is to provide finer-grained supervision for each token during update while maintaining consistency in the optimality with action-level optimization, i.e. maximizing agents' cumulative returns. For arbitrary subsets of actions $w^{1:j}$ with $j \leq |a|$, we define token value functions for supervising policy update as

$$Q_\pi(s, w^{1:j-1}, w^j) \triangleq \mathbb{E}\big[ R^\gamma | s_0 = s, w_0^{1:j-1} = w^{1:j-1}, w_0^j = w^j \big], \tag{7}$$

$$V_\pi(s, w^{1:j-1}) \triangleq \mathbb{E}\big[ R^\gamma | s_0 = s, w_0^{1:j-1} = w^{1:j-1} \big]. \tag{8}$$

A natural approach to approximate $Q_\pi(s, w^{1:j-1}, w^j)$ and $V_\pi(s, w^{1:j-1})$ is conceptualizing the token generation process as part of the MDP, where each token is treated as a micro action. This enables

back-propagating credit among all tokens to furnish detailed supervision. Such an idea is borrowed from the modeling process of RLHF in general language tasks[10, 42]. In this way, the token-level Bellman backup corresponding to Equations 4 and 5 can be expressed as

$$Q_\pi(s_t, w_t^{1:j-1}, w_t^j) \leftarrow \begin{cases} 0 + \gamma_w \max_{w_t^{j+1}} Q_\pi(s_t, w_t^{1:j}, w_t^{j+1}), & \text{if } j < |a_t| \\ R(s_t, a_t) + \gamma_a \max_{w_{t+1}^1} Q_\pi(s_{t+1}, w_{t+1}^1), & \text{if } j = |a_t| \end{cases}, \tag{9}$$

$$V_\pi(s_t, w_t^{1:j}) \leftarrow \begin{cases} 0 + \gamma_w V_\pi(s_t, w_t^{1:j+1}), & \text{if } j < |a_t| \\ R(s_t, a_t) + \gamma_a V_\pi(s_{t+1}, \emptyset), & \text{if } j = |a_t| \end{cases}. \tag{10}$$

To facilitate subsequent theoretical analysis, we separate the discount factor $\gamma$ into intra-action $\gamma_w$ and inter-action $\gamma_a$, despite their numerical equivalence here. The above backups can be interpreted as applying RL algorithms on a modified reward function, which maintains action-level feedback while introducing extra 0 feedback for tokens within an action, except for the last token. Intuitively, this approach seems feasible and decomposes the action-level reward signal $R(s_t, a_t)$ to intra-action tokens, thus alleviating the uncertainty in credit assignment [31] and reducing optimization complexity. However, it must be noted that the optimization of Equations 9 and 10 are inconsistent with those of Equations 4 and 5 for sequential decision-making tasks since it introduces a new MDP $\bar{\mathcal{M}}$ that diverges from the origin one. We will analyze this discrepancy in the following section.

## 4.2 The Discrepancy

To maintain the consistency between action-level updates and token-level updates, we should ensure that optimizing over tokens gives the same optimality as optimizing the whole action, which is analogous to the multi-dimensional action optimization setting in traditional RL[34]. That is, given the deterministic nature of linguistic action generation, and an optimal polity $\pi^*$ after sufficient training following the token-level Bellman backups, ideal optimal token value functions should satisfy $Q_{\pi^*}(s_t, w_t^{1:j-1}, w_t^j) = Q_{\pi^*}(s_t, a_t)$ and $V_{\pi^*}(s_t, w_t^{1:j}) = V_{\pi^*}(s_t)$ for $\forall j < |a_t|$. To quantify the discrepancy between action-level optimization and token-level optimization with the shape of Equations 9 and 10, we expand the value backup process over each token starting from arbitrary $j < |a_t|$ as the following equations (For derivation see Appendix A).

$$Q_{\pi^*}(s_t, w_t^{1:j-1}, w_t^j) = \underbrace{R(s_t, a_t) + \gamma_a \max_{a_t} Q_{\pi^*}(s_{t+1}, a_{t+1})}_{Q_{\pi^*}(s_t, a_t)} \tag{11}$$
$$- \underbrace{\left[ (1 - \gamma_w^{|a_t|-j}) R(s_t, a_t) + \gamma_a (1 - \gamma_w^{|a_t|+|a_{t+1}|-j-1}) \max_{a_{t+1}} Q_{\pi^*}(s_{t+1}, a_{t+1}) \right]}_{\text{Discrepancy between Equation 4 and 9}},$$

$$V_{\pi^*}(s_t, w_t^{1:j}) = \underbrace{R(s_t, a_t) + \gamma_a V_{\pi^*}(s_{t+1})}_{V_{\pi^*}(s_t)} - \underbrace{\left[ (1 - \gamma_w^{|a_t|-j}) R(s_t, a_t) + \gamma_a (1 - \gamma_w^{|a_t|-j}) V_{\pi^*}(s_{t+1}) \right]}_{\text{Discrepancy between Equation 5 and 10}}, \tag{12}$$

With Equation 11 and 12, we observe following significant insights:

(i) *The discrepancy of the Bellman optimality between action-level optimization and token-level optimization diminishes as $\gamma_w \in [0, 1]$ increases, achieving consistency when $\gamma_w = 1$.*

(ii) *Given $\gamma_w < 1$, the discrepancy increases as the number of intra-action tokens, $|a_t|$, increases.*

Based on the first insight, we will derive our main approach, namely the Bellman backup with Action-Decomposition, in the next section. The second insight can be intuitively understood as follows: The diverged MDP $\bar{\mathcal{M}}$ attaches the assumption that "*words later in a sentence are more expressive than earlier ones*", which is unrealistic for representing the semantics of linguistic actions.

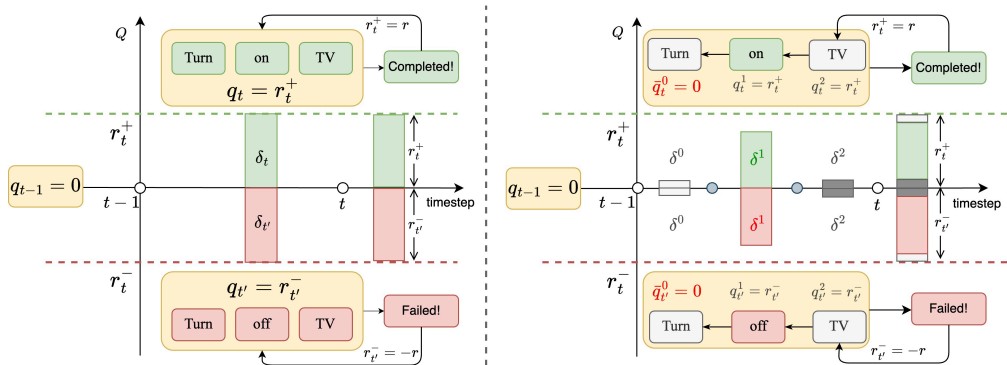

Figure 2: Visual comparison of the differences between action-level Bellman backup (left) and our BAD (right), given the goal *turn on the TV*, where $q$ is the action or token value estimations, $\delta_t = q_t - q_{t-1}$ or $\delta^j = q^j - q^{j-1}$ represent the credit assigned to corresponding actions and tokens respectively for policy update, e.g. the advantage value[43]. To facilitate understanding, a step-by-step breakdown of the right figure is provided in Appendix L.

## 5 Action-Decomposition Reinforcement Learning

### 5.1 Bellman Backup with Action-Decomposition

According to the first insight, we can modify Equations 9 and 10, proposing the *Bellman backup with Action-Decomposition (BAD)* as

$$Q_\pi(s_t, w_t^{1:j-1}, w_t^j) \leftarrow \begin{cases} \max_{w_t^{j+1}} Q_\pi(s_t, w_t^{1:j}, w_t^{j+1}), & \text{if } j < |a_t| \\ R(s_t, a_t) + \gamma \max_{w_{t+1}^1} Q_\pi(s_{t+1}, w_{t+1}^1), & \text{if } j = |a_t| \end{cases}, \quad (13)$$

$$V_\pi(s_t, w_t^{1:j}) \leftarrow \begin{cases} V_\pi(s_t, w_t^{1:j+1}), & \text{if } j < |a_t| \\ R(s_t, a_t) + \gamma V_\pi(s_{t+1}, \emptyset), & \text{if } j = |a_t| \end{cases}. \quad (14)$$

(For proof of optimization consistency see Appendix B.) Training language agents with BAD provides finer-grained supervision for credit backpropagation, eliminating uncertainty in credit assignment and thus enjoying better interpretability and efficiency of the RL training process. In addition, it theoretically ensures consistency between the token-level training process for language models and the RL objective of maximizing actions' utilities. Figure 2 visually compares the differences between action-level Bellman backup and our BAD and demonstrates how BAD precisely assigns credit to each token.

Another advantage of BAD is the ability to decompose the optimization in an intractable action space of size $O(|V|^{|a|})$, into $|a|$ times optimizations in token spaces of size $O(|V|)$, reducing the complexity of RL problem with language agent to $O(|a| \times |V|)$, thus rendering the problem more manageable. Moreover, BAD can be seamlessly integrated into various existing RL methods, including off-policy algorithms e.g. DQN[44], on-policy ones like Actor-Critic[45] and PPO[15]. Furthermore, we have also provided a version of the Soft Q-function in Appendix B.3 to support various entropy-regularized RL algorithms like SAC[46, 47].

### 5.2 Policy Optimization with Action Decomposition

In this section, we integrate the BAD into the widely used PPO and propose a specific method called *Policy Optimization with Action Decomposition (POAD)*, while the integration with other algorithms will be reserved for future works. POAD sequentially decomposes the policy update granularity from the action level to the token level. To approximate the token value function, we introduce a critic network with parameters $\phi$ whose objective is to minimize the empirical Bellman error of tokens by

$$L(\theta) = \frac{1}{T} \sum_{t=0}^{T-1} \left[ \frac{1}{|a_t|} \left( \underbrace{[R(s_t, a_t) + \gamma V_{\bar{\theta}}(s_{t+1}, \emptyset) - V_\theta(s_t, w_t^{1:|a_t|})]^2}_{\text{Inter-action credit assignment}} + \underbrace{\sum_{j=1}^{|a_t|-1} [V_{\bar{\theta}}(s_t, w_t^{1:j+1}) - V_\theta(s_t, w_t^{1:j})]^2}_{\text{Intra-action credit assignment}} \right) \right], \quad (15)$$

where $\bar{\theta}$ represents the non-differentiable parameter of the target network, updated at intervals. The policy network's parameters are denoted as $\phi$, and optimization follows the clipping PPO objective.

$$L(\phi) = -\frac{1}{T} \sum_{t=0}^{T-1} \frac{1}{|a_t|} \sum_{j=1}^{|a_t|} \left[ \min \left( \text{ratio}_t^j(\phi) \hat{A}_t^j, \text{clip}(\text{ratio}_t^j(\phi), 1 \pm \epsilon) \hat{A}_t^j \right) \right], \qquad (16)$$

$$\text{ratio}_t^j(\phi) = \frac{\pi_\phi(w_t^j | s_t, w_t^{1:j-1})}{\pi_{\phi_{old}}(w_t^j | s_t, w_t^{1:j-1})}, \text{ and } \hat{A}_t^j = \hat{A}(w_t^j | s_t, w_t^{1:j-1}),$$

where $\hat{A}_t^j$ is an estimate of the advantage value for each token with the *generalized advantage estimation* (GAE) [48]. To capture more details about POAD, we draw a pseudo-code in Appendix D.

# 6   Experiments

In this section, we show the superiority of POAD in performance, efficiency, and generalization abilities with different testbeds. Moreover, we conduct meaningful ablations on $\gamma_a$ and $\gamma_w$ to verify the theoretical analysis in Section 4.2. Finally, we examine models trained with POAD and baseline methods to investigate their impact on the model's original language abilities. For in-depth analysis, we conduct a case study in Appendix F to validate the effectiveness of BAD in terms of token-level credit assignment. We deploy LLaMA2-7B [49] for Overcooked and VirtualHome, and CodeLLaMA-7b [50] for DataSciCoding, fine-tuned with Low Rank Adaptation (LoRA) [51] with 1 Nvidia A100 GPU.

## 6.1   Environmental Setup

We first evaluate our method on two classical sequential decision-making environments with limited action space: Overcooked [5] and VirtualHome [5], where the action space consists of approximately 10 possible actions per state, each includes 5-10 tokens. Then we evaluate our method in a data science coding and debugging environment with unrestricted action space: DataSciCoding, where agents generate actions (up to 128 tokens) freely. More detailed descriptions can be found in Appendix E.

**Overcooked and VirtualHome.** Overcooked challenges agents to prepare dishes such as *tomato salad* and *tomato-lettuce salad* in a 7x7 grid kitchen using linguistic actions like *Chop, Get-Tomato, and Go-Cutting-Board*, with rewards for correct deliveries and penalties for incorrect ones or time wastage. Meanwhile, VirtualHome simulates household tasks like reheating *pancakes* in Food Preparation and organizing an evening of Entertainment, with actions such as *walk to the living room* and *turn on the TV*, rewarding agents only upon task completion in a partially observable setting.

**DataSciCoding.** We develop DataSciCoding to automate data science coding tasks with unrestricted action space, currently adopting 3 Kaggle datasets and 3 OpenML datasets [52] with details in Appendix E.1. In each task, agents aim to implement the most effective classifier with the scikit-learn module, striving to achieve the highest possible ROC AUC score [53] on test sets. As the prompts provided to the agents contain no detailed information about task datasets, agents are required to interactively modify and debug their code based on feedback from the runtime environment until it works, thus aligning the task datasets. Agents receive ROC AUC scores $\in [0, 1]$ as rewards for workable codes and $-1$ as penalties for run failed. Adopting the same evaluation metrics as CAAFE [54], for each dataset and code, we evaluate 5 repetitions, each with a random $50\% - 50\%$ train-test split [55], and record the average ROC AUC score across these splits.

## 6.2   Baseline Methods

For Overcooked and VirtualHome, we compare POAD's performance with Naive Token-Level Policy Optimization (NTPO) mentioned in Section 4.1, i.e. integrating Equation 10 with $\gamma_w = \gamma_a$ into PPO, and TWOSOME [5]—the current state-of-the-art method on Overcooked and VirtualHome. Besides, we also incorporate ArCHer [26] as a baseline in VirtualHome with comparative analysis, since it is positioned as an intermediate between NTPO and POAD for token-level credit assignment, theoretically enjoying reduced discrepancy compared to NTPO. We demonstrate the difference in the backup processes between POAD and these baselines as well as the optimality after convergence in Appendix C

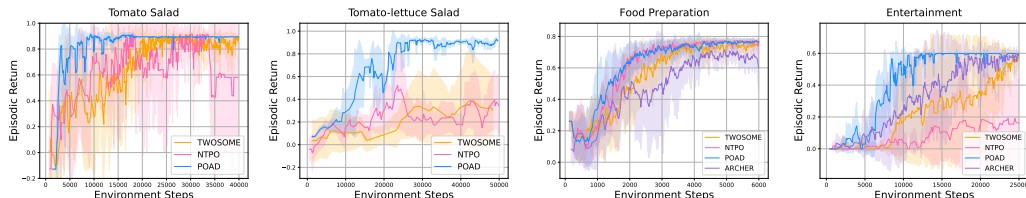

Figure 3: Performance comparisons on Overcooked (first two) and VirtualHome (last two).

For the DataSciCoding environment, in case TWOSOME is inapplicable due to the unrestricted action space, we examine POAD's performance along with NTPO. Meanwhile, we also compare POAD with CAAFE [54]—a novel AutoML framework in which humans collaborate with large language models for data science. In CAAFE, humans implement machine learning models, e.g. classifiers, while large language models generate the code for feature engineering. CAAFE represents the current state-of-the-art performance with collaboration between humans and powerful closed-source LLMs such as GPT-3.5 and GPT-4 on the given task datasets we used.

## 6.3 Main Results

### 6.3.1 Classical Sequential Decision-Making Tasks with Restricted Action Space

As shown in Figure 3, where curves are averaged over 3 seeds with the shadow showing their standard deviation, the performance drop of NTPO when comparing with POAD verifies the existence and negative impact of the discrepancy analyzed in Section 4.2. While POAD can achieve the same (or even better) convergence performance compared to TWOSOME which further verifies the consistency between token-level optimization with BAD and action-level optimization. In addition, the training curves of POAD are more stable (enjoying smaller standard deviations) than TWOSOME and converge much faster than all other baselines, indicating POAD's stability and efficiency by integrating our BAD. In complex Entertainment tasks, ArCHer is second only to POAD, aligning with our theoretical expectations. However, in tasks such as Food Preparation where the methods' performance gap was less pronounced, ArCHer performed poorly, potentially due to instability in its system that involved multiple value networks.

### 6.3.2 Data Science Coding Tasks with Unrestricted Action Space

According to the training curves in Figure 4, our method POAD significantly outperforms NTPO both in the convergence speed and the final score. Compared to the results on sequential decision-making tasks in Section 6.3.1, the performance gap between POAD and NTPO on DataSciCoding is consistently larger, due to the much longer action length $|a_t|$, at most 128 tokens for each action in DataSciCoding. Such results are consistent with our second insights in Section 4.2 and empirically highlight the importance of a proper intra-action credit assignment, which, our Bellman Backup with Action Decomposition (BAD) is designed for.

In Figure 4, POAD-Best means the performance of the best code discovered during POAD's training process with CodeLLaMA-7B. We compare it with the best performance achieved by the state-of-the-art AutoML framework CAAFE [54] with GPT-4 model. In this experiment, we aim to prove that even small-scale language models can also provide better outcomes than large-scale models, what is needed is just a stable and efficient training algorithm POAD and only 2-3 hours on Nvidia A100 (Details of wall-time on each task are shown in Appendix H). A more detailed Comparison between POAD and CAAFE with both GPT-3.5 and GPT-4 can be found in Appendix G.

## 6.4 Open-Vocabulary Task Generalization

LLMs' open-vocabulary feature enables language agents to transfer their learned skills into unseen similar tasks, expanding the capabilities of decision-making agents. We compare the generalization performance of language agents trained by POAD, TWOSOME and NTPO in the *Food Preparation* task with the original base model LLaMA2-7B. Table 1 shows that token-level policy optimization methods achieve better generalization performance in unseen tasks. And our POAD outperforms the other baselines in seven of eight tasks.

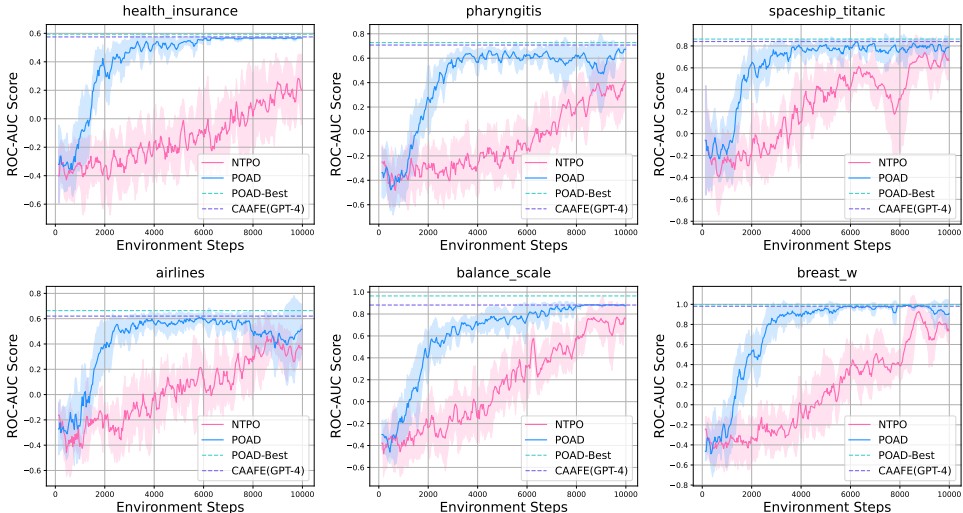

Figure 4: While TWOSOME does not support open action space tasks, we compare the average performance between POAD and NTPO on the DataSciCoding benchmarks, as well as POAD-Best the performance of best code explored by POAD during the training phase and CAAFE with GPT-4.

Table 1: Comparison of generalization performance on eight unseen tasks, with episodic returns averaged over 100 episodes. In these tasks, we replace the *pancake* in the original Food Preparation task with other foods like *cheese*, *hamburger*, *apple pie* and *pizza*, or replace the (*pancake*, *microwave*) at the same time with (*dishes*, *dishwasher*) or (*clothes*, *washing machine*) for greater differences.

| Methods | Cheese | Hamburger | Apple Pie | Pizza | Washing Plate | Laundry |
|---------|--------|-----------|-----------|-------|---------------|---------|
| LLaMA2-7B | 0.1351 | 0.1342 | 0.1656 | 0.1409 | 0.0527 | 0.0344 |
| TWOSOME | 0.7119 | 0.7058 | 0.7304 | 0.7047 | 0.7031 | 0.6038 |
| NTPO | 0.7428 | 0.7476 | 0.7141 | 0.7355 | **0.7491** | 0.5687 |
| POAD | **0.7553** | **0.7602** | **0.7650** | **0.7625** | 0.7075 | **0.7014** |

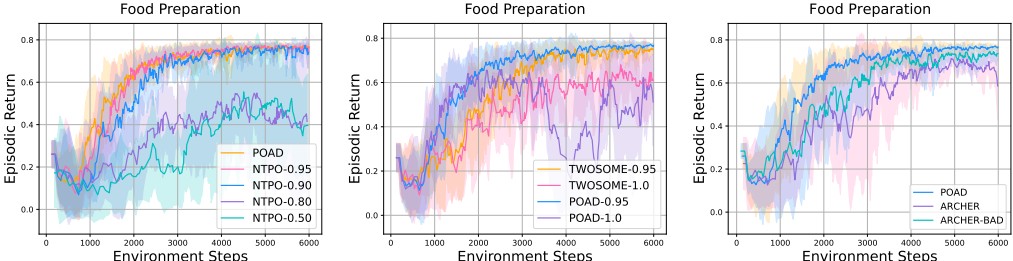

Figure 5: Ablation on $\gamma_a \in \{1.0, 0.95\}$ for both TWOSOME and POAD (left), and $\gamma_w \in \{0.95, 0.9, 0.8, 0.5\}$ for NTPO while keeping the $\gamma_a = 0.95$ unchanged (middle). In the left figure, Setting $\gamma_a = 1.0$ led to decreased performance and convergence for TWOSOME and POAD, validating necessity of $\gamma_a < 1.0$. While in the right figure, the increasingly larger performance gap between POAD and NTPO, as $\gamma_w$ decreases, verifies the theoretical analysis in Section 4.2. The right one shows the performance change after applying theoretical insights to enhance ArCHer's performance, i.e. ARCHER-BAD with $\gamma_w = 1.0$.

## 6.5 Ablations: the Impact of $\gamma_a$ and $\gamma_w$ to the Discrepancy

To verify our analysis in Section 4.2, we conduct ablations on $\gamma_w$ to further investigate how large the discrepancy would be by using different values of $\gamma_w$ in NTPO. Therefore, we deploy NTPO on the *Food Preparation* task with $\gamma_w \in \{0.95, 0.9, 0.8, 0.5\}$, while we keep $\gamma_a = 0.95$ which is consistent with our main experiments. Besides, we also show the performance with $\gamma_a \in \{1.0, 0.95\}$ to show the necessity of setting $\gamma_a$ strictly less than 1.0, and thus the necessity to separate inter-action tokens and intra-action tokens. Further, our theoretical analysis is also compatible with ArCHer, motivating us to apply insights in Section 4.2 to enhance ArCHer's performance.

The left side of Figure 5 demonstrates an increasingly larger performance gap between POAD ($\gamma_w = 1$) and NTPO as $\gamma_w$ decreases. These results empirically showcase the discrepancy between naive token-level credit assignment and our BAD, which is consistent with our first theoretical insight in Section 4.2. Moreover, in the middle of Figure 5, for both TWOSOME and our proposed method POAD, setting $\gamma_a = 1.0$ performs much worse than $\gamma_a = 0.95$, indicating that the discrepancy between NTPO and POAD can not be solved by simply setting $\gamma_a = \gamma_w = 1.0$. Furthermore, the right one in Figure 2 shows improved results that validate the effectiveness of applying our insights to ArCHer. However, due to the inherent challenges of excessive network complexity and difficult hyper-parameter tuning, ArCHer-BAD still falls short of matching POAD's performance.

### 6.6 Impact on Original Language Ability

To investigate the impacts of online RL fine-tuning on LLMs' original capabilities, we evaluate the models trained by POAD, TWOSOME and NTPO on widely used NLP benchmarks[56] which are also reported in Tan et al. [5] and Touvron et al. [49]. These models are trained in *Food Preparation*. Table 2 demonstrates these models' zero-shot performance, compared

Table 2: Zero-shot performance on Language Model Evaluation Harness [56], with details in Appendix J.

| Methods | ARC_C | HellaSwag | PIQA | MMLU |
|---------|-------|-----------|------|------|
| LLaMA2-7B | 0.44 | 0.57 | 0.78 | 0.41 |
| TWOSOME | 0.44 | 0.58 | 0.78 | 0.41 |
| NTPO | 0.44 | 0.58 | 0.78 | 0.41 |
| POAD | 0.45 | 0.59 | 0.78 | 0.41 |

with the original LLaMA2-7B model. The results show no significant losses of general ability like natural language understanding after aligning with the embodied environment, even sometimes bringing minor improvements.

## 7 Conclusion

In this work, we propose the Bellman backup with Action Decomposition (BAD), theoretically eliminating the discrepancy between naive token-level policy optimization and action-level policy optimization for language agents. Integrating BAD with PPO, we propose our method of Policy Optimization with Action Decomposition (POAD), providing finer-grained supervision for each intra-action token and ensuring theoretical consistency between the token-level training nature of language models and the RL objective of maximizing actions' utilities. Empirical experiments and thorough ablations showcase the effectiveness of BAD as well as the superiority of POAD in learning efficiency and generalization abilities, over strong action-level baseline TWOSOME.

**Limitation and Future Work.** The existing limitation of POAD is on the requirement for a quantitative reward function, which is not easily attainable in some environments. To mitigate this, we envisage integrating POAD with self-rewarding [57, 58] or hindsight relabeling [59].

**Social Impact.** The advancements in RL for language agents can significantly enhance decision-making processes in various domains such as healthcare, finance, and autonomous systems. Improved decision-making can lead to better outcomes, increased efficiency, and reduced errors. However, we acknowledge that when optimizing agents using our method, language agents may potentially resort to unscrupulous means to maximize rewards, which could lead to potentially harmful results. Thus, we advocate for a more comprehensive consideration when designing the reward function, or combining it with safety-constrained RL methods to mitigate these risks.

## Acknowledgment

The SJTU team is partially supported by National Key R&D Program of China (2022ZD0114804), Shanghai Municipal Science and Technology Major Project (2021SHZDZX0102) and National Natural Science Foundation of China (62322603, 62076161, 62106141). Muning Wen is supported by Wu Wen Jun Honorary Scholarship, AI Institute, Shanghai Jiao Tong University.

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

# A Derivation for the Discrepancy

## A.1 Q-Function

The optimization over each token following Equation 9, starting from arbitrary $j < |a_t|$, could be expanded as

$$
\begin{aligned}
Q_{\pi^*}(s_t, w_t^{1:j-1}, w_t^j) &= \gamma_w \max_{w_t^{j+1}} Q_{\pi^*}(s_t, w_t^{1:j}, w_t^{j+1}) \qquad\qquad (17)\\
&= \gamma_w \max_{w_t^{j+1}} \left[ \gamma_w \max_{w_t^{j+2}} Q_{\pi^*}(s_t, w_t^{1:j}, w_t^{j+1}, w_t^{j+2}) \right]\\
&= \gamma_w^{|a_t|-j} R(s_t, a_t) + \gamma_a \gamma_w^{|a_t|-j} \max_{w_{t+1}^1} Q_{\pi^*}(s_{t+1}, w_{t+1}^1)\\
&= \gamma_w^{|a_t|-j} \left[ R(s_t, a_t) + \gamma_a \max_{w_{t+1}^1} \left( \gamma_w \max_{w_{t+1}^2} Q_{\pi^*}(s_{t+1}, w_{t+1}^1, w_{t+1}^2) \right) \right]\\
&= \gamma_w^{|a_t|-j} \left[ R(s_t, a_t) + \gamma_a \gamma_w \left( \max_{w_{t+1}^{1:2}} Q_{\pi^*}(s_{t+1}, w_{t+1}^1, w_{t+1}^2) \right) \right]\\
&= \gamma_w^{|a_t|-j} \left[ R(s_t, a_t) + \gamma_a \gamma_w^{|a_{t+1}|-1} \max_{w_{t+1}^{1:|a_{t+1}|}} Q_{\pi^*}(s_{t+1}, w_{t+1}^{1:|a_{t+1}|}) \right]\\
&= \gamma_w^{|a_t|-j} \left[ R(s_t, a_t) + \gamma_a \gamma_w^{|a_{t+1}|-1} \max_{a_{t+1}} Q_{\pi^*}(s_{t+1}, a_{t+1}) \right]\\
&= \underbrace{R(s_t, a_t) + \gamma_a \max_{a_{t+1}} Q_{\pi^*}(s_{t+1}, a_{t+1})}_{Q_{\pi^*}(s_t, a_t)}\\
&\quad - \underbrace{\left[ (1 - \gamma_w^{|a_t|-j}) R(s_t, a_t) + \gamma_a (1 - \gamma_w^{|a_t|+|a_{t+1}|-j-1}) \max_{a_{t+1}} Q_{\pi^*}(s_{t+1}, a_{t+1}) \right]}_{\text{Discrepancy between Equation 4 and 9}}.
\end{aligned}
$$

## A.2 V-Function

Similar to Appendix A.1, the optimization over each token following Equation 10, starting from arbitrary $j < |a_t|$, could be expanded as

$$
\begin{aligned}
V_{\pi^*}(s_t, w_t^{1:j-1}) &= \gamma_w V_{\pi^*}(s_t, w_t^{1:j}) \qquad\qquad (18)\\
&= \gamma_w \left[ \gamma_w V_{\pi^*}(s_t, w_t^{1:j+1}) \right]\\
&= \gamma_w^{|a_t|-j} R(s_t, a_t) + \gamma_a \gamma_w^{|a_t|-j} V_{\pi^*}(s_{t+1})\\
&= \underbrace{R(s_t, a_t) + \gamma_a V_{\pi^*}(s_{t+1})}_{V_{\pi^*}(s_t)} - \underbrace{\left[ (1 - \gamma_w^{|a_t|-j}) R(s_t, a_t) + \gamma_a (1 - \gamma_w^{|a_t|-j}) V_{\pi^*}(s_{t+1}) \right]}_{\text{Discrepancy between Equation 5 and 10}}.
\end{aligned}
$$

# B Proof of Optimization Consistency

To show that optimizing value functions with BAD still ensures the same optimality with action-level optimization, we show that optimizing the value functions for each token is equivalent to optimizing for the full action.

## B.1 Q Function

The optimization over each token with the optimal policy $\pi^*$ following Equation 13, starting from arbitrary $j < |a_t|$, could be expanded as

$$
\begin{aligned}
Q_{\pi^*}(s_t, w_t^{1:j-1}, w_t^j) &= \max_{w_t^{j+1}} Q_{\pi^*}(s_t, w_t^{1:j}, w_t^{j+1}) \\
&= \max_{w_t^{j+1}} \left[ \max_{w_t^{j+2}} Q_{\pi^*}(s_t, w_t^{1:j}, w_t^{j+1}, w_t^{j+2}) \right] \\
&= R(s_t, a_t) + \gamma \max_{w_{t+1}^1} Q_{\pi^*}(s_{t+1}, w_{t+1}^1) \\
&= R(s_t, a_t) + \gamma \max_{w_{t+1}^1} \left( \max_{w_{t+1}^2} Q_{\pi^*}(s_{t+1}, w_{t+1}^1, w_{t+1}^2) \right) \\
&= R(s_t, a_t) + \gamma \left( \max_{w_{t+1}^{1:2}} Q_{\pi^*}(s_{t+1}, w_{t+1}^1, w_{t+1}^2) \right) \\
&= R(s_t, a_t) + \gamma \max_{w_{t+1}^{1:|a_{t+1}|}} Q_{\pi^*}(s_{t+1}, w_{t+1}^{1:|a_{t+1}|}) \\
&= R(s_t, a_t) + \gamma \max_{a_t} Q_{\pi^*}(s_{t+1}, a_{t+1}) \\
&= Q_{\pi^*}(s_t, a_t). 
\end{aligned} \tag{19}
$$

## B.2 V Function

The optimization over each token following Equation 14, starting from arbitrary $j < |a_t|$, could be expanded as

$$
\begin{aligned}
V_{\pi^*}(s_t, w_t^{1:j-1}) &= V_{\pi^*}(s_t, w_t^{1:j}) \\
&= V_{\pi^*}(s_t, w_t^{1:j+1}) \\
&= R(s_t, a_t) + \gamma V_{\pi^*}(s_{t+1}). \\
&= V_{\pi^*}(s_t), 
\end{aligned} \tag{20}
$$

where $V_{\pi^*}(s_t, w_t^{1:j-1})$ and $V_{\pi^*}(s_t, w_t^{1:|a_t|})$ are equivalent to $\mathbb{E}_{w_t^j \sim \pi^*}[Q(s_t, w_t^{1:j-1}, w_t^j)]$ and $\mathbb{E}_{w_{t+1}^1 \sim \pi^*}[Q(s_{t+1}, w_{t+1}^1)]$.

## B.3 Soft Q function in Entropy-Regularized RL

As an extension of BAD, we also provide the soft Bellman backup with Action-Decomposition (sBAD), to support Entropy-Regularized methods like SAC (by setting the reference policy $\bar{\pi}$ to a uniform distribution), as

$$
Q_\pi(s_t, w_t^{1:j-1}, w_t^j) = \begin{cases} \mathbb{E}_{w_t^{j+1} \sim \pi}[Q_\pi(s_t, w_t^{1:j}, w_t^{j+1})] - \beta D_{KL}[\pi||\bar{\pi}](s_t, w_t^{1:j}), & \text{if } j < |a_t| \\ R(s_t, a_t) + \gamma(\mathbb{E}_{w_{t+1}^1 \sim \pi}[Q_\pi(s_{t+1}, w_{t+1}^1)] - \beta D_{KL}[\pi||\bar{\pi}](s_{t+1})), & \text{if } j = |a_t| \end{cases}. \tag{21}
$$

We show that optimizing the soft Q-function with sBAD at the token level is consistent with optimizing for the full action. We adopt $\text{KL}(a|s) = D_{KL}[\pi^*||\bar{\pi}](s)$, where the $\text{KL}(a|s)$ indicate it is a action level KL divergence, $\text{KL}(w|s)$ indicate a token level KL divergence. Given an optimal stochastic policy $\pi^*$, the vanilla soft Bellman backup for full actions is:

$$
\begin{aligned}
\mathbb{E}_{a_t \sim \pi^*}[Q(s_t, a_t)] &= \mathbb{E}_{a_t \sim \pi^*}\left[ R(s_t, a_t) + \gamma(\mathbb{E}_{a_{t+1} \sim \pi^*}[Q(s_t, a_t)] - \beta \text{KL}(a_{t+1}|s_{t+1})) \right] \tag{22} \\
&= \mathbb{E}_{a_t \sim \pi^*}[R(s_t, a_t)] + \gamma \mathbb{E}_{a_t, a_{t+1} \sim \pi^*}[Q(s_{t+1}, a_{t+1})] - \gamma \beta \mathbb{E}_{a_t \sim \pi^*}[\text{KL}(a_{t+1}|s_{t+1})].
\end{aligned}
$$

Following our sBAD, the update over each token within an action ($j < |a|$) is:

$$
\begin{aligned}
\mathbb{E}_{w_t^1 \sim \pi^*}[Q(s_t, w_t^1)] &= \mathbb{E}_{w_t^1 \sim \pi^*}[\mathbb{E}_{w_t^2 \sim \pi^*}[Q(s_t, w_t^1, w_t^2)] - \beta \mathrm{KL}(w_t^2|s_t, w_t^1)] \\
&= \mathbb{E}_{w_t^1, w_t^2 \sim \pi^*}[Q(s_t, w_t^1, w_t^2)] - \beta \mathbb{E}_{w_t^1 \sim \pi^*}[\mathrm{KL}(w_t^2|s_t, w_t^1)] \\
&= \mathbb{E}_{w_t^1, \ldots, w_t^{|a|} \sim \pi^*}[Q(s_t, w_t^{1:|a_t|-1}, w_t^{|a_t|})] \\
&\quad - \beta \mathbb{E}_{w_t^1, \ldots, w_t^{|a_t|-1} \sim \pi^*}\Big[ \sum_{j=1}^{|a_t|-1} \mathrm{KL}(w_t^{j+1}|s_t, w_t^{1:j}) \Big] \\
&= \mathbb{E}_{a_t \sim \pi^*}[Q(s_t, a_t)] \\
&\quad - \beta \mathbb{E}_{w_t^1, \ldots, w_t^{|a_t|-1} \sim \pi^*}\Big[ \sum_{j=1}^{|a_t|-1} \mathrm{KL}(w_t^{j+1}|s_t, w_t^{1:j}) \Big]
\end{aligned}
\tag{23}
$$

Minus $\beta \mathrm{KL}(w_t^1|s_t)$ from both sides, we have:

$$
\mathbb{E}_{w_t^1 \sim \pi^*(s_t)}[Q(s_t, w_t^1)] - \beta \mathrm{KL}(w_t^1|s_t) = \mathbb{E}_{a_t \sim \pi^*}[Q(s_t, a_t)] - \beta \mathrm{KL}(a_t|s_t),
\tag{24}
$$

where $\mathrm{KL}(a_t|s_t) = \sum_{j=1}^{|a_t|} \mathrm{KL}(w_t^j|s_t, w_t^{1:j-1})$.

Then, the update across the current action $a_t$ and the next action $a_{t+1}$ with our sBAD:

$$
\begin{aligned}
\mathbb{E}_{a_t \sim \pi^*}[Q(s_t, a_t)] &= \mathbb{E}_{a_t \sim \pi^*}[R(s_t, a_t)] + \gamma(\mathbb{E}_{a_t \sim \pi^*, w_{t+1}^1 \sim \pi^*}[Q(s_{t+1}, w_{t+1}^1)] \\
&\quad - \beta \mathbb{E}_{a_t \sim \pi^*}[\mathrm{KL}(w_{t+1}^1|s_{t+1})]) \\
&= \mathbb{E}_{a_t \sim \pi^*}[R(s_t, a_t)] + \gamma \mathbb{E}_{a_t, a_{t+1} \sim \pi^*}[Q(s_{t+1}, a_{t+1})] \\
&\quad - \gamma \beta \mathbb{E}_{a_t \sim \pi^*}[\mathrm{KL}(a_{t+1}|s_{t+1})].
\end{aligned}
\tag{25}
$$

Equation 25 enjoys the same shape as Equation 22, thus optimizing the soft Q function following sBAD is equivalent to the action-level optimization. We prove the consistency between sBAD and the traditional soft Bellman updates over full actions.

## C  Comparison of the Optimality between Three Backup Approaches

We provided a visual comparison of the differences between four different backup approaches with the optimal policy after convergence in Figure 6.

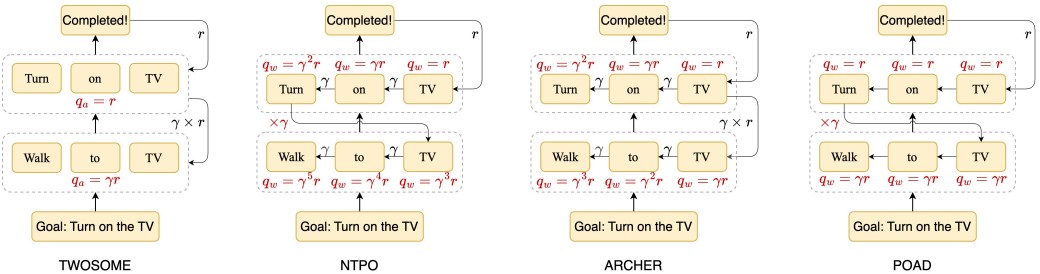

Figure 6: Visual comparison of the differences between four different backup approaches with the optimal policy after convergence. We show that optimizing the Q-function for each token with BAD (POAD) is equivalent to backup for the full action (TWOSOME), while others are not.

## D    Pseudo Code of POAD

---

**Algorithm 1** Policy Optimization with Action Decomposition

---

1: **Input:** LLM $\rho$, Learning rate $\alpha$, MDP $\mathcal{M} = (\mathcal{V}, \mathcal{S}, \mathcal{A}, \mathcal{T}, R, \gamma)$
2: **Initialize:** $\pi_\theta : \{\rho, \theta\}; V_\phi : \{\rho, \phi\}; \bar{\theta} \leftarrow \theta$
3: **Initialize:** Data buffer $\mathcal{D} \leftarrow \emptyset$
4: **for** each epoch **do**
5:     **for** $t = 0$ **to** $T - 1$ **do**
6:         Collect $s_t$.
7:         $a_t \sim \pi_\phi(\cdot|s_t)$.
8:         $s_{t+1} \sim \mathcal{T}(\cdot|s_t, a_t)$.
9:         $\mathcal{D} \leftarrow \mathcal{D} \cup \{(s_t, a_t, R(s_t, a_t), s_{t+1})\}$.
10:     **end for**
11:     Sample a mini-batch $\mathcal{B}$ from $\mathcal{D}$.
12:     **for** each time step $t$ in $\mathcal{B}$ **do**
13:         **for** each token $j$ in $a_t$ **do**
14:             **if** $j < |a_t|$ **then**
15:                 $v_{targ} \leftarrow V_{\bar{\theta}}(s_t, w_t^{1:j+1})$
16:             **else if** $j = |a_t|$ **then**
17:                 $v_{targ} \leftarrow R(s_t, a_t) + \gamma V_{\bar{\theta}}(s_{t+1}, \emptyset)$
18:             **end if**
19:             $v \leftarrow V_\theta(s_t, w_t^{1:j})$
20:             Estimate $\hat{A}_t^j$ with GAE or $v_{targ} - v$
21:         **end for**
22:     **end for**
23:     $\theta \leftarrow \theta - \alpha\nabla_\theta\mathbb{E}_\mathcal{B}[(v - v_{targ})^2]$
24:     $\phi \leftarrow \phi - \alpha\nabla_\phi\mathbb{E}_\mathcal{B}[L(\phi)]$
25:     $\bar{\theta} \leftarrow \theta$
26: **end for**

---

## E    Detailed Description of Experimental Environments

Figure 7 visually shows the Overcooked and VirtualHome tasks.

**Overcooked** is proposed as a typical deicision-making environment in Tan et al. [5]. There are two tasks in which a language agent is aiming to make and serve a *tomato salad* and *tomato-lettuce salad* respectively, with the provided ingredients and tools in a 7×7 grid world as a kitchen. To solve the tasks, the agent needs to explore and learn the correct order to cook the dish with the provided macro-actions, such as *Chop, Get-Tomato, and Go-Cutting-Board*. The environment is partially observable. The agent only observes the objects within 5×5 square centered on the agent. The reward involves +0.2 for chopping a correct ingredient, +1 terminal reward for delivering the correct dish, -0.1 for delivering any wrong item, and -0.001 for every time step.

**VirtualHome** is a simulated physical household environment proposed in Tan et al. [5]. In this environment, an agent is placed in a fully-furnished household with various objects. There are two tasks in the environment, the first one is to find the cold *pancake* on the table and heat it with the *microwave* in the kitchen. In the generalization tasks, we replace the *pancake* with other foods like *hamburger* and *pizza*, or replace the (*pancake*, *microwave*) at the same time with (*dishes*, *dishwasher*) or (*clothes*, *washing machine*), to evaluate agents' generalization abilities for similar but unseen tasks. The second one is to plan to have some entertainment while watching TV, for example picking up chips and milk in the kitchen, bringing them to the living room, turning on the TV, sitting on the sofa and enjoying. In order to solve the tasks, the agent need to use macro-actions to interact with the environment such as *walk to the living room*, *turn on the TV* and *sit on the sofa*. The environment is partially observable. Both tasks adopt a sparse reward setting, only when the task is finished, will the agent receive +1 reward.

**DataSciCoding** is an environment we developed for data science coding tasks with open action space. We adopt 3 Kaggle datasets and 3 OpenML datasets [52] with details in Appendix E.1. In each task,

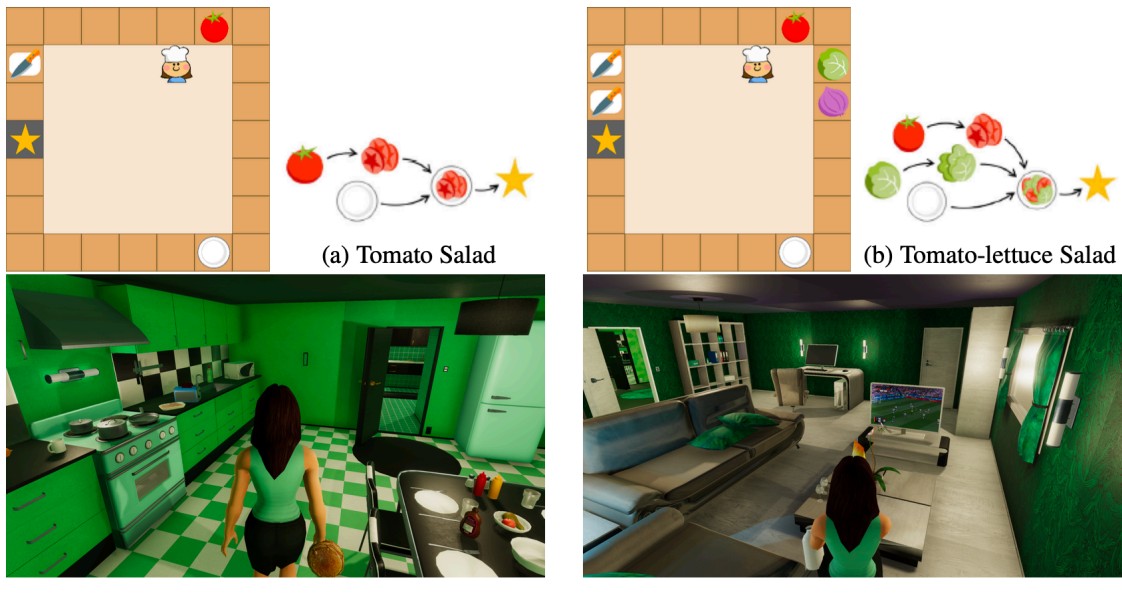

(a) Tomato Salad

(b) Tomato-lettuce Salad

(c) Food Preparation

(d) Entertainment

Figure 7: Visual demonstrations of Overcooked and VirtualHome tasks [5].

agents aim to build the most effective classifier with the scikit-learn module, striving to achieve the highest possible ROC AUC score [53] on test sets. As the prompts provided to the agents contain no detailed information about task datasets, agents are required to align their knowledge to different datasets, that is, they must continuously modify and debug their code based on feedback from the runtime environment until it works. Therefore, it is a sequential decision-making process. Through this interaction, they align their approach to the specific task dataset. When the codes are executed successfully, a done signal will occur immediately, along with an ROC AUC score $\in [0, 1]$. If the codes run wrong, the agent receives a constant $-1.0$ as a penalty. Adopting the same evaluation metrics as Hollmann et al. [54], for each dataset, we evaluate 5 repetitions, each with a random $50\% - 50\%$ train-test split [55], and record the average ROC AUC score across these splits.

### E.1 Details of Datasets Used in DataSciCoding Tasks

These datasets are collected from Kaggle and OpenML respectively, with pruning approaches consistent with Hollmann et al. [54].

Table 3: Details of task datasets in DataSciCoding, where [K] denotes Kaggle.

| Data set | #Features | #Samples | #Classes |
|---|---|---|---|
| Pharyngitis [K] | 19 | 512 | 2 |
| Health Insurance [K] | 13 | 2000 | 2 |
| Spaceship Titanic [K] | 13 | 2000 | 2 |
| Airlines | 7 | 2000 | 2 |
| Balance Scale | 4 | 125 | 3 |
| Breast-w | 9 | 69 | 2 |

## F Case Study in Token-Level Credit Assignment

In this section, we conduct a case study for in-depth analysis to validate the effectiveness of BAD in terms of token-level credit assignment. We selected the last three states from the Food Prepa-

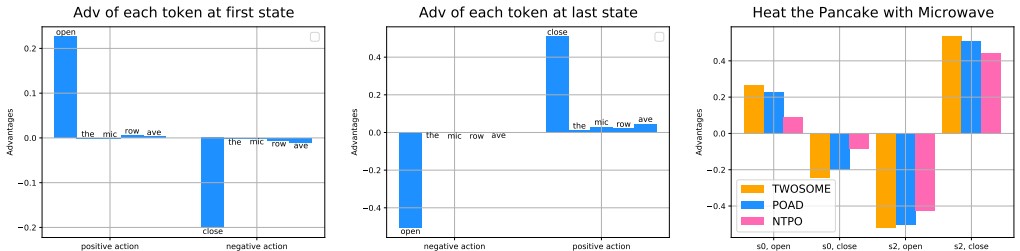

Figure 8: Case Study: Demonstration of the token-level credit assignment learned by the BAD at two states (left two). And a comparison of the volume of credit assignment for key tokens between different methods (right one), where TWOSOME indicates the credit assigned to entire actions instead of specific tokens.

ration task, which form a complete subtask: *Heat the Pancake with Microwave*. This serves as a simple yet practical case for our analysis, with transitions: $\mathcal{T}(s_0, \text{"open the microwave"}) \rightarrow s_1$, $\mathcal{T}(s_0, \text{"close the microwave"}) \rightarrow s_0$, $\mathcal{T}(s_1, \text{"put the pancake into the microwave"}) \rightarrow s_2$, $\mathcal{T}(s_2, \text{"open the microwave"}) \rightarrow s_2$, $\mathcal{T}(s_2, \text{"close the microwave"}) \rightarrow$ success with reward 1.0. According to these transitions, the optimal trajectory to complete this subtask is (*open the microwave, put the pancake into the microwave, close the microwave*). Additionally, the maximum step length for the task is 5; if the task is not completed within 5 steps, it is considered a failure, resulting in a reward signal of -1.0.

Based on this, we first sample 1,000 examples using a random policy, then train a critic to convergence with three different backups: action-level Bellman backup (TWOSOME), naive token-level Bellman backup (NTPO), and BAD (POAD). In Figure 3, we recorded the credit assignment results with each critic for each token in positive and negative actions under the first and last states.

The first two in Figure 8 show the credit assigned to each token in a symmetric (positive, negative) action pair by the BAD-learned critic in two different states, results confirm that the BAD critic can effectively assign most of the credit to the key tokens while rarely influencing other irrelevant tokens. The right one illustrates the volume of credits assigned to key tokens by POAD and NTPO, compared with the credit assigned to the entire action by TWOSOME, showing that POAD enjoys a far smaller discrepancy than NTPO.

## G  Detailed Comparison between POAD and CAAFE

Table 4 shows the performance of the best code discovered during POAD's training process with CodeLLaMA-7B, comparing with CAAFE with GPT-3.5 and GPT-4.

Table 4: The performance of the best code discovered during POAD's training process with CodeLLaMA-7B, comparing with CAAFE with GPT-3.5 and GPT-4. [$K$] denotes that the dataset is collected from Kaggle, while others are collected from OpenML.

| Task | POAD-Best | CAAFE(GPT-3.5) | CAAFE(GPT-4) |
|---|---|---|---|
| health-insurance[$K$] | **0.5939**±0.01 | 0.5745±0.02 | 0.5748±0.02 |
| pharyngitis[$K$] | **0.7282**±0.01 | 0.6976±0.03 | 0.7078±0.04 |
| spaceship-titanic[$K$] | **0.8628**±0.01 | 0.8383±0.02 | 0.8405±0.02 |
| airlines | **0.664**±0.01 | 0.619±0.04 | 0.6203±0.04 |
| balance-scale | **0.9651**±0.03 | 0.844±0.31 | 0.882±0.26 |
| breast-w | **0.9981**±0.01 | 0.9809±0.02 | 0.9809±0.02 |

## H  Wall-time of POAD on DataSciCoding Environment

Table 5 shows the average wall-time of training LLaMA2-7b with LoRA and POAD on all DataSci-Doing tasks.

Table 5: Average wall-time for POAD training with each dataset with 1 * Nvidia A100.

| Task | Wall-Time | Environmental Steps |
|------|-----------|---------------------|
| health-insurance[$K$] | 2h 34m | 10k |
| pharyngitis[$K$] | 2h 11m | 10k |
| spaceship-titanic[$K$] | 3h 5m | 10k |
| airlines | 2h 56m | 10k |
| balance-scale | 1h 43m | 10k |
| breast-w | 2h 6m | 10k |

# I   Details in Generalization Experiments

Table 6: Comparison of generalization performance on eight unseen tasks, with episodic returns averaged over 100 episodes (The fewer timesteps consumed, the higher the returns). In these tasks, we replace the *pancake* in the original Food Preparation task with other foods like *cheese*, *hamburger*, *apple pie* and *pizza*, or replace the (*pancake*, *microwave*) at the same time with (*dishes*, *dishwasher*) or (*clothes*, *washing machine*) for greater differences. In parentheses is the success rate of completing the task within 50 timesteps.

| Methods | Cheese | Hamburger | Apple Pie | Pizza | Washing Plate | Laundry |
|---------|--------|-----------|-----------|-------|---------------|---------|
| LLaMA2-7B | 0.1351(0.55) | 0.1342(0.55) | 0.1656(0.61) | 0.1409(0.55) | 0.0527(0.27) | 0.0344(0.15) |
| TWOSOME | 0.7119(1.0) | 0.7058(1.0) | 0.7304(1.0) | 0.7047(1.0) | 0.7031(1.0) | 0.6038(1.0) |
| NTPO | 0.7428(1.0) | 0.7476(1.0) | 0.7141(1.0) | 0.7355(1.0) | **0.7491(1.0)** | 0.5687(1.0) |
| POAD | **0.7553(1.0)** | **0.7602(1.0)** | **0.7650(1.0)** | **0.7625(1.0)** | 0.7075(1.0) | **0.7014(1.0)** |

# J   Evaluation on NLP Benchmarks

To investigate the impacts of fine-tuning different RL methods on the LLM's original abilities, we evaluate the models trained by POAD, NTPO and TWOSOME on widely used NLP benchmarks[56]. All models are trained in the Food Preparation task. The four benchmarks are ARC_Challenge, HellaSwag, PIQA and MMLU, with results on ARC_Challenge, HellaSwag and PIQA are displayed in Table 7 and the results of MMLU are displayed in Table 8. All results are calculated following the default configurations in Gao et al. [56].

Table 7: Zero-shot performance on Common Sense Reasoning tasks

| Methods | ARC_C | HellaSwag | PIQA |
|---------|-------|-----------|------|
| LLaMA2-7B | 0.4352 | 0.5713 | 0.7807 |
| TWOSOME | 0.4445 | 0.5819 | 0.7786 |
| NTPO | 0.4428 | 0.5825 | 0.7824 |
| POAD | 0.4471 | 0.5856 | 0.7797 |

Table 8: Details of Zero-shot performance on Massive Multitask Language Understanding tasks

| Tasks | LLaMA2-7B | TWOSOME | NTPO | POAD |
|---|---|---|---|---|
| abstract_algebra | 0.3100 | 0.2900 | 0.3300 | 0.2900 |
| anatomy | 0.4296 | 0.4296 | 0.4444 | 0.4593 |
| astronomy | 0.4474 | 0.4145 | 0.4342 | 0.4079 |
| business_ethics | 0.4300 | 0.4300 | 0.4400 | 0.4500 |
| clinical_knowledge | 0.4868 | 0.4566 | 0.4792 | 0.4604 |
| college_biology | 0.4236 | 0.4097 | 0.4236 | 0.4306 |
| college_chemistry | 0.2900 | 0.3000 | 0.2700 | 0.2700 |
| college_computer_science | 0.2900 | 0.3000 | 0.3100 | 0.3000 |
| college_mathematics | 0.3700 | 0.3900 | 0.4000 | 0.3900 |
| college_medicine | 0.4451 | 0.4104 | 0.4220 | 0.4393 |
| college_physics | 0.2451 | 0.1961 | 0.2157 | 0.2157 |
| computer_security | 0.5400 | 0.5100 | 0.5300 | 0.5100 |
| conceptual_physics | 0.3532 | 0.3660 | 0.3489 | 0.3489 |
| econometrics | 0.2193 | 0.2105 | 0.2018 | 0.1930 |
| electrical_engineering | 0.3724 | 0.3586 | 0.3931 | 0.3655 |
| elementary_mathematics | 0.2460 | 0.2619 | 0.2672 | 0.2593 |
| formal_logic | 0.3413 | 0.3571 | 0.3175 | 0.3175 |
| global_facts | 0.3100 | 0.2900 | 0.3400 | 0.2800 |
| high_school_biology | 0.4516 | 0.4548 | 0.4452 | 0.4548 |
| high_school_chemistry | 0.3300 | 0.3054 | 0.3350 | 0.3153 |
| high_school_computer_science | 0.3800 | 0.4000 | 0.4100 | 0.4100 |
| high_school_european_history | 0.6000 | 0.5879 | 0.5636 | 0.5879 |
| high_school_geography | 0.4343 | 0.4545 | 0.4444 | 0.4495 |
| high_school_government_and_politics | 0.5389 | 0.5181 | 0.5233 | 0.5181 |
| high_school_macroeconomics | 0.3769 | 0.3718 | 0.3821 | 0.3821 |
| high_school_mathematics | 0.2704 | 0.2593 | 0.2481 | 0.2630 |
| high_school_microeconomics | 0.3739 | 0.3739 | 0.3655 | 0.3487 |
| high_school_physics | 0.2781 | 0.3179 | 0.2848 | 0.2914 |
| high_school_psychology | 0.5248 | 0.5376 | 0.5138 | 0.5174 |
| high_school_statistics | 0.2731 | 0.2824 | 0.2778 | 0.2546 |
| high_school_us_history | 0.5196 | 0.5441 | 0.5343 | 0.5049 |
| high_school_world_history | 0.5612 | 0.5823 | 0.5696 | 0.5696 |
| human_aging | 0.4350 | 0.4484 | 0.4395 | 0.4260 |
| human_sexuality | 0.5649 | 0.5267 | 0.5344 | 0.5191 |
| international_law | 0.5620 | 0.5785 | 0.5620 | 0.5702 |
| jurisprudence | 0.4722 | 0.4722 | 0.4630 | 0.4537 |
| logical_fallacies | 0.4785 | 0.4663 | 0.4663 | 0.4847 |
| machine_learning | 0.3929 | 0.3839 | 0.3571 | 0.3571 |
| management | 0.4466 | 0.4272 | 0.4272 | 0.4175 |
| marketing | 0.6026 | 0.6282 | 0.6239 | 0.5940 |
| medical_genetics | 0.4900 | 0.5000 | 0.5000 | 0.4800 |
| miscellaneous | 0.5428 | 0.5428 | 0.5581 | 0.5504 |
| moral_disputes | 0.4480 | 0.4249 | 0.4451 | 0.4364 |
| moral_scenarios | 0.2402 | 0.2413 | 0.2559 | 0.2425 |
| nutrition | 0.4771 | 0.4869 | 0.4739 | 0.4739 |
| philosophy | 0.4887 | 0.4695 | 0.4952 | 0.4823 |
| prehistory | 0.4660 | 0.4568 | 0.4784 | 0.4691 |
| professional_accounting | 0.3546 | 0.3511 | 0.3617 | 0.3511 |
| professional_law | 0.3096 | 0.3181 | 0.3136 | 0.3123 |
| professional_medicine | 0.4118 | 0.4228 | 0.3713 | 0.4044 |
| professional_psychology | 0.4199 | 0.4248 | 0.4167 | 0.4085 |
| public_relations | 0.4182 | 0.4636 | 0.4273 | 0.4364 |
| security_studies | 0.4735 | 0.4694 | 0.4571 | 0.4857 |
| sociology | 0.6020 | 0.6119 | 0.6269 | 0.6169 |
| us_foreign_policy | 0.6700 | 0.6400 | 0.6500 | 0.6400 |
| virology | 0.4217 | 0.4337 | 0.4157 | 0.4036 |
| world_religions | 0.6140 | 0.6023 | 0.5965 | 0.5965 |

# K   Hyper-Parameters Settings of Experiments

During experiments, the implementations of TWOSOME are consistent with their official repositories, reproducing the original best-performing status. We first list the hyper-parameter candidates used for grid search in Table 9. Then, we show the hyper-parameters that achieve the best performance for each method and environment in Table 10-17.

Table 9: Hyper-Parameters candidates for grid search in Overcooked, VirtualHome, and DataSciCoding environments.

| hyper-parameters | candidates |
|---|---|
| critic lr | 1e-3,5e-4,1e-4,5e-5,1e-5, |
| actor lr | 5e-4,1e-4,5e-5,1e-5,5e-6,1e-6,5e-7 |
| ppo epochs | 1,5 |
| num mini batch | 2,4,8,16,32 |
| gamma | 0.99,0.95 |
| entropy coef | 0.01,0.001,0.0001 |
| max grad norm | 10,0.5 |

Table 10: Hyper-parameters used for POAD in Overcooked tasks.

| hyper-parameters | value | hyper-parameters | value | hyper-parameters | value |
|---|---|---|---|---|---|
| critic lr | 1e-5 | actor lr | 5e-7 | ppo epochs | 5 |
| batch size | 128 | num mini batch | 2 | gamma | 0.99 |
| rollout threads | 4 | entropy coef | 0.01 | max grad norm | 0.5 |
| PPO clip | 0.2 | value coef | 0.5 | KL threshold | 0.02 |

Table 11: Hyper-parameters used for NTPO in Overcooked tasks.

| hyper-parameters | value | hyper-parameters | value | hyper-parameters | value |
|---|---|---|---|---|---|
| critic lr | 1e-5 | actor lr | 5e-7 | ppo epochs | 5 |
| batch size | 128 | num mini batch | 4 | gamma | 0.99 |
| rollout threads | 4 | entropy coef | 0.01 | max grad norm | 0.5 |
| PPO clip | 0.2 | value coef | 0.5 | KL threshold | 0.02 |

Table 12: Hyper-parameters used for TWOSOME in Overcooked tasks.

| hyper-parameters | value | hyper-parameters | value | hyper-parameters | value |
|---|---|---|---|---|---|
| critic lr | 1e-5 | actor lr | 5e-7 | ppo epochs | 1 |
| batch size | 128 | num mini batch | 4 | gamma | 0.99 |
| rollout threads | 4 | entropy coef | 0.01 | max grad norm | 0.5 |
| PPO clip | 0.2 | value coef | 0.5 | KL threshold | 0.02 |

Table 13: Hyper-parameters used for POAD in VirtualHome tasks.

| hyper-parameters | value | hyper-parameters | value | hyper-parameters | value |
|---|---|---|---|---|---|
| critic lr | 5e-5 | actor lr | 1e-6 | ppo epochs | 5 |
| batch size | 128 | num mini batch | 2 | gamma | 0.95 |
| rollout threads | 4 | entropy coef | 0.0001 | max grad norm | 0.5 |
| PPO clip | 0.2 | value coef | 0.5 | KL threshold | 0.02 |

Table 14: Hyper-parameters used for NTPO in VirtualHome tasks.

| hyper-parameters | value | hyper-parameters | value | hyper-parameters | value |
|---|---|---|---|---|---|
| critic lr | 5e-5 | actor lr | 1e-6 | ppo epochs | 5 |
| batch size | 128 | num mini batch | 2 | gamma | 0.95 |
| rollout threads | 4 | entropy coef | 0.01 | max grad norm | 0.5 |
| PPO clip | 0.2 | value coef | 0.5 | KL threshold | 0.02 |

Table 15: Hyper-parameters used for TWOSOME in VirtualHome tasks.

| hyper-parameters | value | hyper-parameters | value | hyper-parameters | value |
|---|---|---|---|---|---|
| critic lr | 5e-5 | actor lr | 1e-6 | ppo epochs | 1 |
| batch size | 128 | num mini batch | 4 | gamma | 0.95 |
| rollout threads | 4 | entropy coef | 0.01 | max grad norm | 0.5 |
| PPO clip | 0.2 | value coef | 0.5 | KL threshold | 0.02 |

Table 16: Hyper-parameters used for POAD in DataSciCoding tasks.

| hyper-parameters | value | hyper-parameters | value | hyper-parameters | value |
|---|---|---|---|---|---|
| critic lr | 5e-5 | actor lr | 1e-6 | ppo epochs | 1 |
| batch size | 128 | num mini batch | 4 | gamma | 0.95 |
| rollout threads | 4 | entropy coef | 0.01 | max grad norm | 0.5 |
| PPO clip | 0.2 | value coef | 0.5 | KL threshold | 0.02 |

Table 17: Hyper-parameters used for NTPO in DataSciCoding tasks.

| hyper-parameters | value | hyper-parameters | value | hyper-parameters | value |
|---|---|---|---|---|---|
| critic lr | 5e-5 | actor lr | 1e-6 | ppo epochs | 1 |
| batch size | 128 | num mini batch | 4 | gamma | 0.95 |
| rollout threads | 4 | entropy coef | 0.01 | max grad norm | 0.5 |
| PPO clip | 0.2 | value coef | 0.5 | KL threshold | 0.02 |

# L  Step-by-Step Breakdown of BAD

In this section, to facilitate the understanding of how BAD precisely assigns credit to each token, a step-by-step breakdown of the credit assignment process with BAD is shown in Figures 9-13

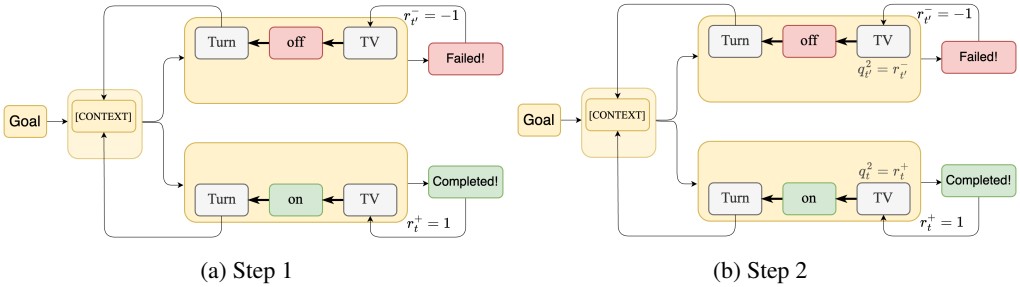

(a) Step 1          (b) Step 2

Figure 9: Step 1: Receiving feedback $r_t^+$ and $r_{t'}^-$ for positive and negative trajectories. Step 2: Propagating $r_t^+$ and $r_{t'}^-$ to $Q(\text{"TV"}|o_t, \text{"Turn on"})$ and $Q(\text{"TV"}|o_t, \text{"Turn off"})$.

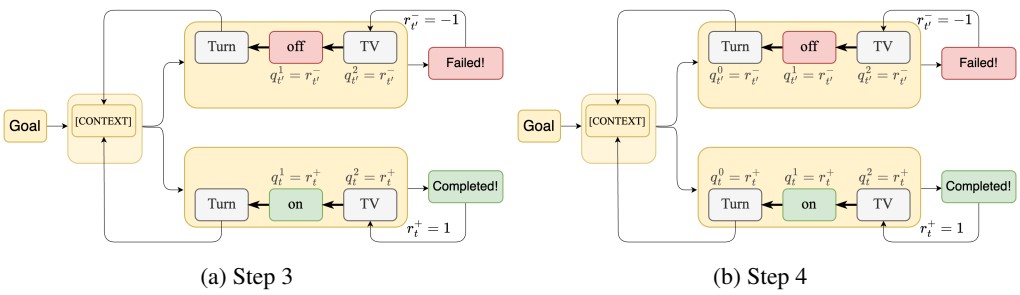

(a) Step 3          (b) Step 4

Figure 10: Step 3 and 4: Back-propagating $r_t^+$ and $r_{t'}^-$ to $Q(\text{"on"}|o_t, \text{"Turn"})$ and $Q(\text{"off"}|o_t, \text{"Turn"})$, and then to $Q(\text{"Turn"}|o_t)$ in both trajectories continuously.

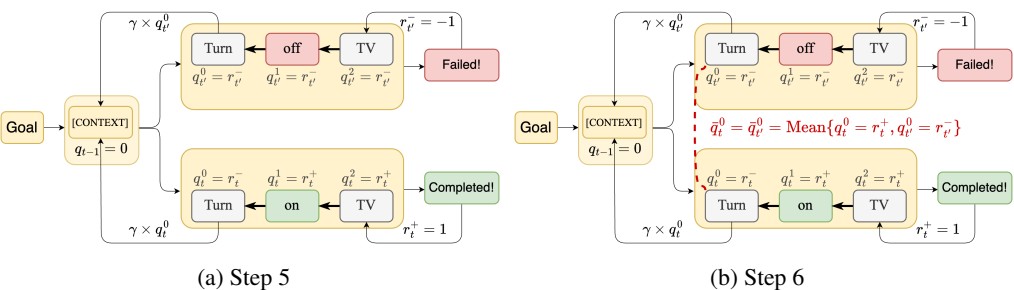

(a) Step 5          (b) Step 6

Figure 11: Step 5: Back-propagating to the previous action with $q_{t-1}^{|a_{t-1}|} = \text{Mean}\{\gamma \times q_t^0, \gamma \times q_{t'}^0\}$ since both trajectories share the same pre-action. Step 6: Similarly, $Q(\text{"Turn"}|o_t)$ is also shared among two trajectories, thus $Q(\text{"Turn"}|o_t) = \text{Mean}\{q_t^0, q_{t'}^0\} = 0$ as well.

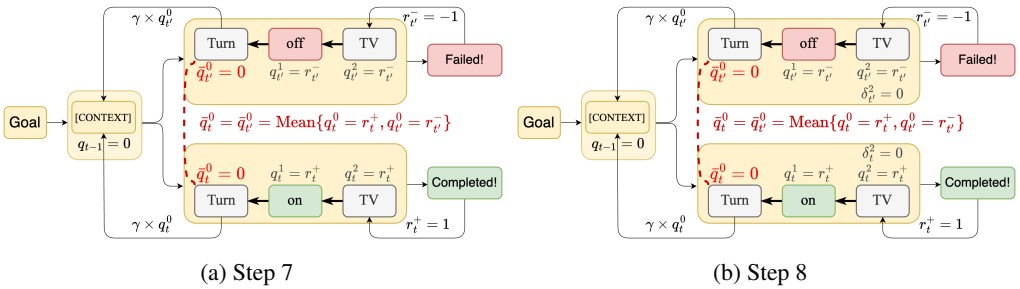

(a) Step 7                                    (b) Step 8

Figure 12: Step 7: Modifying $\bar{q}_t^0$ and $\bar{q}_{t'}^0$ in both trajectories with the same $Q(\text{"Turn"}|o_t)$. Step 8: Starting to back-propagate again for credits assigned to each token for optimization with $\delta^j = q^j - q^{j-1}$ for $j > 0$; this process is similar to the calculation for advantage values.

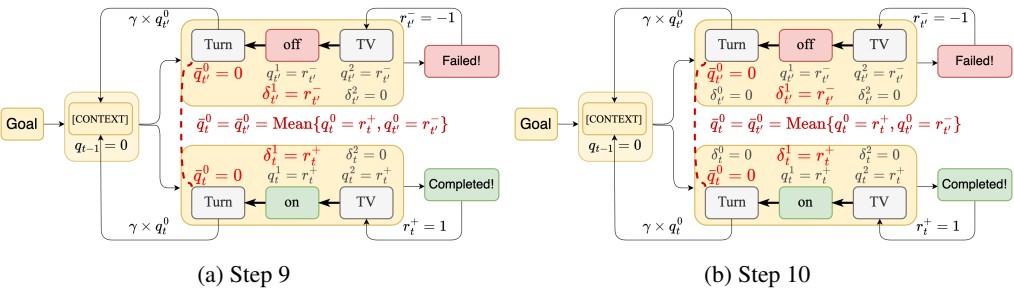

(a) Step 9                                    (b) Step 10

Figure 13: Step 9: Calculating the credits for key tokens, where $\delta_t^1 = q_t^1 - q_t^0$ and $\delta_{t'}^1 = q_{t'}^1 - q_{t'}^0$. Step 10: Calculating the credits for tokens $j = 0$ with $\delta_t^0 = \bar{q}_t^0 - q_{t-1}^{|a_{t-1}|}$ and $\delta_{t'}^0 = \bar{q}_{t'}^0 - q_{t-1}^{|a_{t-1}|}$. Till now, we precisely emphasized key tokens while keeping irrelevant tokens with 0 credits.

