# OpenReview forum: "Reinforcing LLM Agents via Policy Optimization with Action Decomposition"
_NeurIPS.cc/2024/Conference — NeurIPS 2024 poster_

### Official Review · Reviewer_9Wzo · 2024-07-05

**Soundness:** 3
**Presentation:** 3
**Contribution:** 3
**Rating:** 6
**Confidence:** 3

**Summary:**

The paper proposes Bellman backup with Action Decomposition (BAD) and its realization on PPO (POAD), which aims to train with token-level policy for more fine-grained credit assignment on the crucial part of the language response made by the language agent. To address the issue of distorted Q-value by the intra-sentence discount factor, the authors propose a novel Bellman backup rule that applies discount factor and reward only at the end of each sentence (high-level action), and provide the realization, POAD, on PPO. Experiments have shown that the proposed method works better than several baselines on decision-making and coding tasks.

**Strengths:**

1. The paper is well-written and easy to follow. The paper does a good job in stating its motivation; Figure 1 clearly shows why token-level training is necessary, while Eq. 11 and 12 in Sec 4.2 helps much in understanding the inherent problem of token-level agent to be addressed in the rest of the paper. The appendix shows a good amount of details that helps the understanding of the paper, including step-by-step breakdown, hyperparameter tables, wall-clock training time and pseudocode.

2. The proposed method is tested on a variety of test cases, including decision-making and coding, which shows the wide range of potential application for the proposed method. There is also ablation showing the generalizability (i.e., not overfitting) of the trained model, which is a desired property of the LLM community.

**Weaknesses:**

While working better than sentence-level method TWOSOME, the paper is still not convincing enough in showing that token-level critic is indeed better than sentence-level critic for three reasons: 1) there are potential disadvantages such as increased training complexity; 2) the paper did not show results that "key token" identified by token-level critic actually matters; 3) there are other sentence-level critic RL methods, and TWOSOME might not be the best sentence-level critic RL method. See questions section for details.

**Minor issues:**

The colors of the lines are inconsistent throughout the paper. For example, in Fig. 5, POAD is orange in the left subfigure, but it is purple and blue in the right subfigure.

**Questions:**

I have a question for the authors: in line 200-202, the author claims that "another advantage of BAD is ... reducing the complexity of RL problem with language agent to $O(|a|\times|V|)$. While it is true that the action space at each step reduces, the episode length is greatly increased from a few steps to potentially hundreds or thousands of steps with much sparser reward, which could be potentially harder. Could the author explain intuitively why the benefit of reduced action space outweighs the problem of extended episode length with sparse reward?

Also, it will be great if the author can do the following two things (which corresponds to point 2) and 3) in weakness):

1. conduct an ablation showing the intuition in Fig. 2 is actually true, i.e., for the sentence "turn on / off TV", the Q-value actually changes more on the word "on / off";

2. compare with ArCHer [1], which is a sentence-level RL method, to further show that token-level Bellman update is better.

**Reference**

[1] Y. Zhou et al. ArCHer: Training Language Model Agents via Hierarchical Multi-Turn RL. In ArXiv, 2024.

**Limitations:**

The authors mentioned the problem of acquiring quantitative reward function as the limitation, which is a valid concern in some LLM tasks such as reasoning with chain-of-thought, and gave reasonable directions for solution. The paper did not mention any societal impact, for which I would encourage the authors to include a paragraph (the paper is not "theoretical"; it proposes empirical solutions with experiments).

---

> ### Author Rebuttal · Authors · 2024-08-06
>
> ### We thank Reviewer 9Wzo for his/her constructive comments that will surely turn our paper into a better shape.
>
> > **Q1** Could the author explain intuitively why the benefit of reduced action space outweighs the problem of extended episode length with sparse reward?
>
> **A1** Thank you for your thoughtful question. We believe this discussion is valuable and will significantly help readers better understand our work. **We plan to add a dedicated discussion section on this topic in the revised version.**
>
> At first glance, action decomposition indeed extends the episode length, seemingly exacerbating the sparse reward problem. However, we can analyze this issue more deeply from different perspectives:
> 1. Efficiency of credit backpropagation. The increased episode length due to action decomposition could lead to inefficient training if using a 1-step TD, as credit would only backpropagate one step per training epoch. Fortunately, BAD can be combined with n-step TD or GAE to achieve efficient propagation comparable to action-level Bellman backup within a single training epoch. For this reason, our POAD implementation is based on GAE. We appreciate the reviewer's reminder and will explicitly state in future versions that BAD is best used in conjunction with GAE or N-step TD methods to avoid confusion and misuse.
> 2. Credit decay. While the extended episode length could cause exponential decay of credit strength based on the discount factor, which makes it difficult for sparse reward signals to act on actions/tokens in the early stages of an episode, BAD removes the intra-action discount factor. This theoretically prevents the exponential decay of signal strength caused by action decomposition. Our experimental results, where POAD significantly outperforms NTPO, confirm this point.
> 3. Credit estimation variance. Even with addressing the efficiency and decay issues, increased horizons often lead to higher variance in credit estimation. However, in linguistic action environments, the dynamics between intra-action tokens are strictly deterministic. In this case, we are reasonable to expect that the additional variance introduced by action decomposition would be much less severe than that in traditional RL problems with increased episode length.
>
> We hope this discussion intuitively explains why the benefit of reduced action space outweighs the problem of extended episode length with sparse reward when using BAD. Besides, our experiments on the VirtualHome also empirically confirm this. When the agent only receives a binary reward signal upon task completion or reaching the maximum step limit (50 steps), our POAD still outperforms the action-level method, TWOSOME, despite such a sparse reward setting.
>
> > **Q2** Conduct an ablation showing the intuition in Fig. 2 is actually true.
>
> **A2** We are very grateful for this insightful suggestion and **we have conducted a case study/ablation for in-depth analysis to validate the effectiveness of BAD**. Please refer to the **Figure 3: Case Study** section in meta-responses for detailed settings and experiemental results.
>
> By comparing the credit assigned to each token in a symmetric (positive, negative) action pair by the BAD-learned critic in different states, our experimental results confirm that the BAD critic can effectively assign most of the credit to the key tokens while rarely influencing other irrelevant tokens. By investigating the volume of credits assigned to key tokens by POAD and NTPO, compared with the credit assigned to the entire action by TWOSOME, results show that the POAD enjoys a far smaller discrepancy than NTPO.
>
> > **Q3** compare with ArCHer to further show that token-level Bellman update is better.
>
> **A3** We are very grateful to the reviewer for recommending this timely work. Although ArCHer [1] adopts sentence-level critics, it also aims to provide fine-grained token-level supervision for optimizing language agents in interactive (multi-turn) tasks. Upon further investigation, we found that its key idea for token-level credit assignment plays an intermediate role between NTPO and POAD. Thus, in our revised manuscript, **we have incorporated ArCHer as a new baseline in our experiments with comparative analysis.**
>
> Please refer to the **Figures 1 and 2: New Baseline, ArCHer and Corresponding Analysis** section in meta-responses for detailed comparison between ArCHer and our methods, as well as preliminary experimental results. Thanks again for the reviewers’ valuable recommendation, we believe these comparisons and analyses involving ArCHer will better illustrate the advantage of our method as well as the research trajectory in the community toward providing finer-grained supervision for language agents.
>
> > **Q4** Minor Issue: The colors of the lines are inconsistent throughout the paper.
>
> **A4** Thanks to the reviewer for his/her careful examination, we have adjusted the related figures and unified the color used in our updated version.
>
> > **Q5** Lack of societal impact.
>
> **A5** We appreciate the reviewer's suggestion. In our updated version, we have added the following paragraph addressing the societal impact:
>
> **Social Impact.** The advancements in RL for language agents can significantly enhance decision-making processes in various domains such as healthcare, finance, and autonomous systems. Improved decision-making can lead to better outcomes, increased efficiency, and reduced errors. However, we acknowledge that when optimizing agents using our method, language agents may potentially resort to unscrupulous means to maximize rewards, which could lead to potentially harmful results. Thus, we advocate for a more comprehensive consideration when designing the reward function, or combining it with safety-constrained RL methods to mitigate these risks.
>
> ---
> [1] Zhou, Yifei, et al. "Archer: Training language model agents via hierarchical multi-turn rl." arXiv preprint arXiv:2402.19446 (2024).

---

> > ### Comment · Reviewer_9Wzo · 2024-08-09
> >
> > Thanks for the detailed response. I think the authors have addressed most of my concern, including empirical comparison with ArCHer, societal impact, and whether Fig. 2 is true. However, there is one problem: is the updated figure for ArCHer in the global pdf correct? Judging from their Fig. 2 in the arXiv version, they never use a token-level critic and thus should not have discount rate inside a sentence. In fact, in page 9 they wrote: "Informally, Theorem 1 shows that the error in estimating advantages using the token-level critic is $\gamma\sqrt{L}$ larger than the utterance-level critic (in the worst case)". Thus, I think ArCHer's figure should be the same as POAD.

---

> ### Author Response · Authors · 2024-08-09
> **Response to Reviewer 9Wzo**
>
> Thank you very much for your recognition of our efforts.
>
> Yes, ArCHer adopts sentence-level critics exclusively and does not utilize token-level critics. However, the absence of token-level critics does not always imply that there is no discount factor applied within sentences. In Section 3.4 of the original ArCHer paper, the part of "Low-level token actor," authors stated: "We update the token-level policy with the policy gradient computed via REINFORCE (also known as the Monte Carlo Policy Gradient)". This indicates that rather than approximating token-level credits with critic networks, they employ Monte Carlo estimation to backpropagate token-level credits from sentence-level advantage values, $A(s_c, a_t)$, which serve as terminal rewards (and thus bypass the need for token-level critic networks).
>
> We also noted that in one instance provided by ArCHer, specifically in Equation (3), they applied a vanilla (relatively old) version [2] of REINFORCE that does not involve a discount factor (they didn't explain why). However, it is important to highlight that in most contemporary practices involving Monte Carlo estimation or REINFORCE, the use of a discount factor is standard for managing horizons and balancing bias-variance (e.g. $G_t=\sum_{k=0}^\infty \gamma^k r_{t+k+1}$, see Equation (13.8) in Section 13.3 and Equation (3.8) in Section 3.3 of [3]). Furthermore, considering that the authors claim ArCHer is a framework that can integrate with other reinforcement learning approaches, we decided to retain the discount factor in the REINFORCE process as depicted in Figure 1 of the global PDF for a more general representation.
>
> Nevertheless, we have considered ArCHer without a discount factor, which aligns with our insights presented in the paper. We conducted an ablation study on this aspect, as illustrated in Figure 2 of the global PDF, i.e. the ArCHer-BAD variant in the right one. In this case, the performance of ArCHer should theoretically match that of POAD, as they share the same theoretical results of credit assignment. Unfortunately, due to the inherent challenges associated with extra network complexity and the difficulties of hyperparameter tuning, ArCHer still falls short of achieving the performance levels of POAD.
>
> Thank you so much again for this valuable feedback, we will add this clarification along with Figure 1 of the global PDF in our later version.
>
> ---
> [2] Williams, Ronald J. "Simple statistical gradient-following algorithms for connectionist reinforcement learning." Machine learning 8 (1992): 229-256.
>
> [3] Sutton, Richard S., and Andrew G. Barto. Reinforcement learning: An introduction. MIT press, 2018.

---

> > ### Comment · Reviewer_9Wzo · 2024-08-10
> >
> > Thanks for the detailed response. I think this is a minor issue anyway, since the proposed method has already been compared with ArCHer-BAD. The ArCHer paper has github repo available; I suggest the authors to double-check the repo to make sure the figure is correct. In conclusion, I decide to keep my score.

---

> > > ### Author Response · Authors · 2024-08-10
> > > **Thank you for your deep engagement**
> > >
> > > We would like to say thank you for your deep engagement with our work, especially the direction you have suggested helps us make the paper more complete and understandable. We will check the ArCHer repo again and refine our statement continuously. Thanks a lot!

---

### Official Review · Reviewer_671a · 2024-07-10

**Soundness:** 2
**Presentation:** 3
**Contribution:** 2
**Rating:** 6
**Confidence:** 3

**Summary:**

The paper proposes a novel approach to optimizing language agents in reinforcement learning (RL) environments, addressing the challenges of limited environmental knowledge and vast action spaces. Traditional methods like GLAM and TWOSOME optimize language actions as whole units, leading to inefficiencies in credit assignment and optimization complexity. This paper introduces Policy Optimization with Action Decomposition (POAD), which decomposes actions to the token level, allowing for finer-grained supervision and manageable optimization complexity. The authors derive the Bellman backup with Action Decomposition (BAD) to ensure theoretical consistency and integrate it with the PPO algorithm. They validate POAD across various testbeds, demonstrating its advantages in learning efficiency, generalization, and theoretical correctness.

**Strengths:**

- This paper introduces a novel method of decomposing actions to the token level, which provides finer-grained supervision and reduces optimization complexity.
- The derivation of the Bellman backup with Action Decomposition (BAD) ensures theoretical consistency, addressing a missing gap in previous research.
- The paper provides many empirical validation across different testbeds, demonstrating the effectiveness and efficiency of POAD compared to baseline methods.

**Weaknesses:**

1. This paper lacks a thorough comparison and analysis of some important pioneering works, such as LLM decoding from a control or reinforcement learning perspective and hierarchical reinforcement learning.
2. This paper emphasizes that BAD can perform efficient and reasonable credit assignment at the token level, but the experimental section lacks in-depth analysis, such as case studies.

**Questions:**

1. Lines 201-203, the authors mention that the action space is reduced from exponential to polynomial compared to works like GLAM and TWOSOME. However, from Equation 16, it can be seen that the algorithm learned by BAD still samples each token from the entire vocabulary, which does not seem to change the action space?
2. Can the authors demonstrate the token-level credit assignment learned by the BAD method through a case study?
3. BAD has a significant correlation with some existing token-level optimization methods [2][3] and hierarchical optimization methods [3]. Can the authors make a more in-depth comparison between BAD and these methods? Is there any equivalence between BAD and these methods? Is it just extending existing methods, such as [2][3], from single-turn to multi-turn?

---

[1] Zhou, Yifei, et al. "Archer: Training language model agents via hierarchical multi-turn rl." arXiv preprint arXiv:2402.19446 (2024).

[2] Rafailov, Rafael, et al. "From $ r $ to $ Q^* $: Your Language Model is Secretly a Q-Function." arXiv preprint arXiv:2404.12358 (2024).

[3] Zeng, Yongcheng, et al. "Token-level Direct Preference Optimization." ICML 2024

**Limitations:**

The author has already discussed the limitations of the method and its potential impacts in the paper, and provided possible solutions.

---

> ### Author Rebuttal · Authors · 2024-08-06
>
> ### We thank Reviewer 671a for his/her constructive comments that will surely turn our paper into a better shape.
>
> > **Q1** BAD has a significant correlation with some existing token-level optimization methods [2][3] and hierarchical optimization methods [1]. Can the authors make a more in-depth comparison between BAD and these methods? Is there any equivalence between BAD and these methods? Is it just extending existing methods, such as [2][3], from single-turn to multi-turn?
>
> **A1** We appreciate the reviewer for recommending these relevant works. Firstly, [2] and [3] primarily focus on single-turn RL problems, performing decomposition and token-level optimization on single-step outputs. While these works, including [3] here, emphasize the importance of fine-grained token-level supervision, they didn’t take policy consistency into account (which has a minor impact on single-turn tasks). Our work, however, concentrates on multi-turn scenarios, where the discrepancy grows with the number of turns, making it an unavoidable challenge in multi-turn RL. Moreover, one of the baselines in our paper - the naive token-level Bellman backup behind NTPO - can be viewed as a direct extension of [2] and [3] from single-turn to multi-turn. BAD, then, is an improvement that further addresses the inconsistency issue. This highlights both the differences and connections between our method and [2][3]. Our experimental results demonstrate that resolving the inconsistency problem, beyond simply extending, has a significant improvement on the ultimate performance of language agents in multi-turn RL problems. **We will incorporate these comparisons in our Related Work section.**
>
> As for ArCHer [1], we are very grateful to the reviewer for recommending this timely work. ArCHer aims to provide fine-grained token-level supervision for optimizing language agents in interactive (multi-turn) tasks. Upon further investigation, we found that its key idea for token-level credit assignment plays an intermediate role between NTPO and POAD. Thus, **in our revised manuscript, we have incorporated ArCHer as a new baseline in our experiments with comparative analysis.**
>
> Please refer to the **Figures 1 and 2: New Baseline, ArCHer and Corresponding Analysis** section in meta-responses for detailed comparison between ArCHer and our methods, as well as preliminary experimental results. Thanks again for the reviewers’ valuable recommendation, we believe these comparisons and analyses involving ArCHer will better illustrate the advantage of our method as well as the research trajectory in the community toward providing finer-grained supervision for language agents.
>
> > **Q2** This paper emphasizes that BAD can perform efficient and reasonable credit assignment at the token level, but the experimental section lacks in-depth analysis, such as case studies. Can the authors demonstrate the token-level credit assignment learned by the BAD method through a case study?
>
> **A2** We are very grateful for this insightful suggestion and **we have conducted a case study in updated manuscript for in-depth analysis to validate the effectiveness of BAD**. Please refer to the **Figure 3: Case Study** section in meta-responses for detailed settings and experiemental results.
>
> By comparing the credit assigned to each token in a symmetric (positive, negative) action pair by the BAD-learned critic in different states, our experimental results confirm that the BAD critic can effectively assign most of the credit to the key tokens while rarely influencing other irrelevant tokens. By investigating the volume of credits assigned to key tokens by POAD and NTPO, compared with the credit assigned to the entire action by TWOSOME, results show that the POAD enjoys a far smaller discrepancy than NTPO.
>
> > **Q3** Lines 201-203, the authors mention that the action space is reduced from exponential to polynomial compared to works like GLAM and TWOSOME. However, from Equation 16, it can be seen that the algorithm learned by BAD still samples each token from the entire vocabulary, which does not seem to change the action space?
>
> **A3** We apologize for the confusion. BAD does not change the action space. In lines 200-203, we intended to convey that, by using BAD, the critic can provide more fine-grained token-level supervision signals, i.e. the token-level advantage values as shown in Equation 16, for the language agent's policy updates, which reduces the complexity of policy optimization from exponential to polynomial. Specifically, before using BAD, we have an action-level signal, which is used to optimize a policy with an action space of $O(|V|^{|a|})$. After using BAD, we have $|a|$ signals, each guiding an optimization with a token space of $O(|V|)$. Essentially, BAD decomposes a large problem into several smaller problems, thereby reducing the complexity of solving it. Thank you for your comment, we acknowledge that our initial statement might have caused some confusion, and **we will further clarify our statement in lines 200-203 based on this discussion to avoid any unnecessary confusion.**
>
> ---
> [1] Zhou, Yifei, et al. "Archer: Training language model agents via hierarchical multi-turn rl." arXiv preprint arXiv:2402.19446 (2024).
>
> [2] Rafailov, Rafael, et al. "From 𝑟 to 𝑄∗: Your Language Model is Secretly a Q-Function." arXiv preprint arXiv:2404.12358 (2024).
>
> [3] Zeng, Yongcheng, et al. "Token-level Direct Preference Optimization." ICML 2024

---

> ### Author Response · Authors · 2024-08-12
> **We kindly inquire if reveiwer has any further comments or concerns regarding our responses**
>
> Dear reviewer 671a,
>
> We sincerely appreciate the time and effort you dedicated to our work and we have made every effort to address your concerns in our rebuttal submission. Given that the discussion deadline is less than 24 hours away, we wanted to kindly inquire if you have any further comments or concerns regarding our responses. Your insights are incredibly important to us, and we are eager to ensure that all of your points have been adequately addressed. If you feel that our responses have addressed your concerns, we would greatly appreciate your consideration in reflecting this in the final evaluation of the manuscript.
>
> Thank you once again for your time and support throughout this process. We look forward to hearing from you soon.
>
> Best regards,
>
> Authors

---

> > ### Comment · Reviewer_671a · 2024-08-12
> >
> > Many thanks to the authors for thoroughly supplementing the key experiments and discussions. I have raised the score to 6.

---

> > > ### Author Response · Authors · 2024-08-12
> > > **Thank you very much for your recognition of our efforts!**
> > >
> > > Thank you very much for your recognition of our efforts. Your constructive feedback has been invaluable in guiding our revisions and enhancing the quality of our work. Thanks a lot!

---

### Official Review · Reviewer_JxkT · 2024-07-11

**Soundness:** 3
**Presentation:** 3
**Contribution:** 2
**Rating:** 6
**Confidence:** 4

**Summary:**

The paper investigates LLM agents: RL agents where sampling actions from a policy means sampling a sequence of tokens mapping to an action from a (suitably conditioned) large language model. Authors notice a problem with previous implementations of this idea: since actions are typically described as a sequence of more than one token, this introduces issues around credit assignment between tokens contributing to an action. However, fixing it naively, treating each token as an individual action, leads to a problem: when time discounting is present in the MDP, the new (intra-token) MDP is no longer equivalent to the old one. The proposed solution is to only discount "fully formed" actions. This idea is then integrated with standard PPO implementation, and empirically tested against SOTA on three RL environments, showing a meaningful improvement. In addition, the paper presents evidence that another fix of simply removing the discount factor (ie setting it to 1) degrades the performance, and through ablation study proves that it is indeed the intra-token discounting removal that improved the final performance.

**Strengths:**

The paper is written clearly, and investigates an important topic. The experiments convinced me that indeed, the method works as described and improves the performance of LLM agents. I also highly appreciated the investigation into removing time discounting completely and the impact of varying intra-token discount factor, and into the possible degradation of NL capability of the model after fine-tuning.

**Weaknesses:**

I am sceptical with regards to the formal treatment of POMDPs in the paper. First, the definition is missing the initial state (/initial distribution). Then, there are numerous issues: the reward function is defined as operating on states and actions, but is then used as taking an observation and an action. Expectations in equations (1) and (2) are ill-defined if conditioned only on observations: they either have to be conditioned on states, or the observation has to somehow induce a probability distribution over states. The definition of Bellman updates for value function update in eq (5) and (10) refer to the next action/next token, which haven't been defined, and also uses $|a_t|$ which is ambiguous for a given sequence $w_t^{1:j}$. The equations (11) and (12) refer to the next observation $o_{t+1}$ which is a random variable (first, because POMDP, second, because the transition function is not assumed to be deterministic).

I understand that the method was implemented and works fine, and the intuition behind it is clear, so I believe that these mistakes are fundamentally not that serious. However, they confuse and frustrate careful reader, and have to be fixed. (In comparison, TWOSOME paper uses simple MDP formalism, while GLAM paper sweeps the formalism of POMDPs under the rug by informally talking about "approximations").

---

Small notes:
 - typo in "policy" in line 173
 - Figure 2 was very hard to read, and referring the reader to the appendix to understand it - which is, I think, necessary - seems to warrant additional work to make it more readable
 - "NTPO" typo in the description of Figure 4
 - LHS and RHS subfigures are swapped in Figure 5

**Questions:**

1. There is a well-known equivalence between time-discounted MDP and adding an auxiliary state $\s_0$ with zero reward and transition probabilities $T(s, s_0) = \gamma$ for every other state $s$. As far as I understand, it fits into the paradigm in the paper by postulating that additional transitions are added from completed actions only - do authors have any additional insight into this?
2. Did authors thought about incorporating the discount factor trick into ReAct? Since the impact of $\gamma$ is rather large (from experimental evidence), it seems that distinguishing between thoughts (potentially incurring low discount penalty) and true env actions might be a potentially useful direction.

**Limitations:**

Yes.

---

> ### Author Rebuttal · Authors · 2024-08-06
>
> ### We thank Reviewer JxkT for his/her constructive comments that will surely turn our paper into a better shape.
>
> > **Q1** The unclear formalism and notation errors.
>
> **A1** We apologize for the confusion caused to the reviewers and readers. Indeed, we recognize that the use of POMDP in our formulation was unnecessary, as our paper does not delve into discussions related to partial observability. The inaccurate use of POMDP and the resulting omissions have inadvertently introduced additional confusion and distraction for the readers. To address this issue, in the updated version of our manuscript, we have:
>
> 1. referring to TWOSOME, replaced the POMDP formulation with MDP and adjust all related notations accordingly. We believe this change will make our formulation and derivation more clear and concise.
> 2. after these modifications, thoroughly reviewed all notations and equations to ensure that all relevant variables are properly defined and clearly explained.
>
> > **Q2** There is a well-known equivalence between time-discounted MDP and adding an auxiliary state  $s_0$ with zero reward and transition probabilities $T(s,s_0)=\gamma$ for every other state $s$. As far as I understand, it fits into the paradigm in the paper by postulating that additional transitions are added from completed actions only. Do authors have any additional insight into this?
>
> **A2** Yes, we agree with the reviewer's insight about the equivalence between adding the additional transitions for only completed actions, i.e., action-level, and our BAD paradigm. Meanwhile, we appreciate the reviewer providing us with another perspective to understand why removing the intra-action discount is necessary. Specifically, in environments with linguistic actions, the dynamics between intra-action tokens within an action are strictly deterministic. That is, given a token $w^j$ for $(s,w^{1:j−1})$ , we can only transition to $(s,w^{1:j})$. In this case, if we do not remove the intra-action discount, it would be equivalent to adding an extra transition for intra-action tokens, which contradicts the deterministic nature of linguistic action generation.
>
> Additionally, we can consider the implications of setting different discount factors for intra-action and inter-action tokens from another perspective. Originally, the discount factor was used to balance short-term and long-term rewards, with a large discount factor typically applied in scenarios where actions within the horizon are strongly correlated. In our work, given the particularity of intra-action tokens—where meaningful environmental feedback can only be obtained if intra-action tokens can be combined into a valid action—it is crucial to encourage agents to generate highly correlated intra-action tokens. Therefore, it is reasonable to consider setting the intra-action discount factor to 1, then agents will evaluate each of its intra-action tokens based on the action-level feedback. This approach aligns with the core idea of solving specific MDPs using time-varying discount factors, as seen in works like [1].
>
>
> > **Q3** Did the authors think about incorporating the discount factor trick into ReAct?
>
> **A3** Yes, we recognize the potential benefits of integrating different discount factors into distinct components of the language agent's outputs, such as actions and thoughts. Based on the following intuitions:
>
> 1. The thoughts do not directly affect the final state-action transition—only the final action does—instead, thoughts guide the generation to converge to specific low-level actions (they play different roles in interactive tasks).
> 2. We can remove the gamma in intra-action tokens since words later in an action may not be more expressive than earlier ones. However, thoughts usually present a progressive structure, i.e. content closer to the final action may be more closely related to the final action, which makes it reasonable to distinguish them from final actions, e.g. with a low discount factor.
>
> For now, we have observed that discount factors significantly influence the final policy of language agents learned from RL algorithms. Thus, considering separate outputs with varying intentions in a ReAct-like agent, we believe applying different discount factors to these components could be beneficial, whether through heuristic methods or learning-based approaches. For instance, [2] proposed a gradient-based meta-learning approach to optimize the discounting ratio of future rewards, while [1] solves specific MDPs using time-varying discount factors. Both could be promising directions to explore in the future.
>
> > **Q4** Figure 2 should be more readable.
>
> **A4** Thanks very much for the reviewer's suggestion. We are working hard to improve the readability of this figure. **Figure 4 (b) in the meta-responses** is a current version of this diagram, hopefully more readable than the previous one. Besides, to facilitate understanding the effects of BAD shown in this figure, we also conducted a case study (See the **Figure 3: Case Study** section in meta-response) to visually demonstrate the token-level credit assignment results learned by the BAD.
>
> > **Q5** Typos and subfigures swapping issue.
>
> **A5** Thanks to the reviewer’s careful examination, we have addressed these typos and adjusted the figures in our updated version.
>
> ---
> [1] Gan, Jiarui, et al. "Markov decision processes with time-varying geometric discounting." Proceedings of the AAAI Conference on Artificial Intelligence. Vol. 37. No. 10. 2023.
>
> [2] Xu, Zhongwen, Hado P. van Hasselt, and David Silver. "Meta-gradient reinforcement learning." Advances in neural information processing systems 31 (2018).

---

> > ### Comment · Reviewer_JxkT · 2024-08-12
> > **Response**
> >
> > I thank the authors for providing a thorough response, and for committing to fixing the issues pointed out in the review. Based on this, but also others reviewers feedback regarding the baselines comparison, and authors providing additional experiments, I decided to increase my score by one point.

---

> > > ### Author Response · Authors · 2024-08-12
> > > **Thank you for the time and effort you dedicated to re-evaluating our work!**
> > >
> > > We sincerely appreciate the time and effort you dedicated to re-evaluating our work. Your constructive feedback has been invaluable in guiding our revisions and enhancing the quality of our work. Thanks a lot!

---

### Official Review · Reviewer_H4uQ · 2024-07-23

**Soundness:** 2
**Presentation:** 2
**Contribution:** 2
**Rating:** 6
**Confidence:** 2

**Summary:**

This paper introduces Policy Optimization with Action Decomposition (POAD), a novel method for reinforcing language agents by optimizing at the token level rather than the action level. The authors derive a theoretical framework called Bellman backup with Action Decomposition (BAD) to address discrepancies between action-level and token-level optimization. Implementing BAD within PPO, POAD offers finer-grained credit assignment and lower optimization complexity. Experiments across various environments demonstrate POAD's improved learning efficiency and generalization compared to baseline methods, supporting the theoretical analysis and advancing language agent optimization in interactive settings.

**Strengths:**

1. The paper seems to provide a theoretical foundation for its approach, analyzing the discrepancies between action-level and token-level optimization and deriving the Bellman backup with Action Decomposition (BAD) to address these issues. This theoretical grounding lends good analysis to the proposed method.

2.  The authors implement their theoretical insights into a practical algorithm (POAD) and demonstrate its effectiveness across diverse testbeds, including environments with both restricted and unrestricted action spaces. The empirical results show improved learning efficiency and generalization abilities compared to baseline methods, validating the practical utility of the theoretical contributions.

**Weaknesses:**

Limited testing environment and lack of baseline. The author only show two baselines for comparison. More baselines should be include for comprehensive analysis.

Another main concern is that the finetuning is only done on using Lora, when finetune the model using the full capacity, the effect is unknown.

**Questions:**

1. How did you tune the baseline?
2. If you use full scale finetuning instead of Lora, what do you expect? Could you perform on experimentation on this?
3. How would this method work on standard NLP task?

**Limitations:**

Yes, author addressed the limitations.

---

> ### Author Rebuttal · Authors · 2024-08-06
>
> ### We thank Reviewer H4uQ for his/her constructive comments that will surely turn our paper into a better shape.
>
> > **Q1** More baselines should be included for comprehensive analysis.
>
> **A1** We appreciate the feedback from H4uQ and also extend our gratitude to Reviewers 671a and 9Wzo for recommending the recent relevant method, ArCHer [1]. ArCHer aims to provide fine-grained token-level supervision for optimizing language agents on interactive (multi-turn) tasks. Upon further investigation, we found that its key idea for token-level credit assignment plays an intermediate role between NTPO and POAD. Thus, **in our revised manuscript, we have incorporated ArCHer as a new baseline in our experiments with comparative analysis.**
>
> Please refer to the **Figures 1 and 2: New Baseline, ArCHer and Corresponding Analysis** section in meta-responses for detailed comparison between ArCHer and our methods, as well as preliminary experimental results. We believe the extra comparisons and analyses involving ArCHer will better illustrate the advantage of our method and the research trajectory in the community toward providing finer-grained supervision for language agents.
>
> > **Q2** If you use full-scale finetuning instead of Lora, what do you expect? Could you experiment with this?
>
> **A2** We conducted an ablation comparing our method with the action-level baseline, TWOSOME, on both full-scale and LoRA fine-tuning settings, with results referring to **Figure 4 (a) in our meta-responses**.
> 1. Comparing POAD's performance under full-scale fine-tuning and LoRA fine-tuning, we observed consistent performance, indicating that POAD remains effective under full-scale fine-tuning. However, due to increased memory consumption with full-scale fine-tuning, we significantly reduced the batch size during training, leading to slightly reduced stability in the training process. This accounts for the observed larger fluctuations in the curves during full-scale fine-tuning compared to LoRA fine-tuning.
> 2. Contrasting the two methods under full-scale fine-tuning, POAD demonstrated greater advantages over the action-level baseline method compared to Lora fine-tuning. Potential reasons for this include: a) full-scale fine-tuning's heightened sensitivity, requiring higher precision and granularity of supervised signals compared to Lora; b) the further increase in demand for precision of supervised signals due to the drastically reduced batch size.
>
> We greatly appreciate the reviewer for this valuable suggestion. These experimental results further underscore the importance of our approach, highlighting that in scenarios involving full-scale fine-tuning, fine-grained and precise supervision signals are more critical compared to situations involving LoRA fine-tuning.
>
> > **Q3** How did you tune the baseline?
>
> **A3** Regarding the baseline, such as TWOSOME, we integrated its implementation into our code framework based on the official repository. Throughout this process, for fair experimental comparisons, we ensured the following:
> 1. Key components of the baseline method aligned with those in the original repository.
> 2. Be able to replicate the performance reported in the paper using the original hyper-parameters list in their papers.
> 3. Regarding the code framework, apart from necessary adaptations specific to the method, we maintained consistency in other aspects.
> 4. When tuning hyperparameters, we utilized the same search ranges to maintain consistency (we reused the search ranges as defined in TWOSOME).
>
> > **Q4** How would this method work on standard NLP tasks?
>
> **A4** Theoretically, our method can also be applied to standard NLP tasks such as QA, writing, coding, etc (DataSciCoding in our experiments is a coding environment where agents can generate code freely). Notably, the most suitable application scenarios would be those language tasks involving multi-turn interactions (which language agents usually target). In multi-turn situations, our method's advantage in decomposing inter-action and intra-action credit assignments can be fully leveraged. For other single-turn tasks, in the worst-case scenario, our method would be similar to standard RL approaches with terminal rewards derived from environmental feedback or a reward model.
>
> ---
> [1] Zhou, Yifei, et al. "Archer: Training language model agents via hierarchical multi-turn rl." arXiv preprint arXiv:2402.19446 (2024).

---

> ### Author Response · Authors · 2024-08-12
> **We kindly inquire if reveiwer has any further comments or concerns regarding our responses**
>
> Dear reviewer H4uQ,
>
> We sincerely appreciate the time and effort you dedicated to our work and we have made every effort to address your concerns in our rebuttal submission. Given that the discussion deadline is less than 24 hours away, we wanted to kindly inquire if you have any further comments or concerns regarding our responses. Your insights are incredibly important to us, and we are eager to ensure that all of your points have been adequately addressed. If you feel that our responses have addressed your concerns, we would greatly appreciate your consideration in reflecting this in the final evaluation of the manuscript.
>
> Thank you once again for your time and support throughout this process. We look forward to hearing from you soon.
>
> Best regards,
>
> Authors

---

> > ### Comment · Reviewer_H4uQ · 2024-08-12
> >
> > thanks the author, I have increased my score

---

> > > ### Author Response · Authors · 2024-08-12
> > > **Thanks for upgrading the score!**
> > >
> > > We would like to say thank you sincerely for your recognition of our efforts. Your constructive feedback has been invaluable in guiding our revisions and enhancing the quality of our work. Thanks a lot!

---

### Author Rebuttal · Authors · 2024-08-06

# Meta Responses
We are delighted to receive positive feedback from all the reviewers, and thank the reviewers for their valuable suggestions.
## Answers to some common questions and New results (see the PDF attached here)
For convenience, we provide detailed answers to some of the common questions here.
### Figures 1 and 2: New Baseline, ArCHer and Corresponding Analysis
We appreciate the feedback from reviewer H4uQ about involving extra baselines and also extend our gratitude to reviewers 671a and 9Wzo for recommending the recent relevant method, ArCHer [1]. Upon further investigation, we found that its key idea for token-level credit assignment plays an intermediate role between NTPO and POAD. Thus, **we have incorporated ArCHer as a new baseline in our experiments with comparative analysis.**

Specifically, from a theoretical perspective: Figure 1 demonstrates the key processes of TWOSOME, NTPO, ArCHer, and POAD, in terms of credit assignment. ArCHer, positioned as an intermediate between NTPO and POAD, theoretically enjoys reduced discrepancy compared to NTPO (although the discrepancy issue is not explicitly discussed in its literature). Importantly, our theoretical analysis regarding BAD is compatible with ArCHer, allowing us to apply insights gained from our theoretical analysis to optimize ArCHer.

From a practical perspective: ArCHer is implemented based on a hierarchical RL framework, utilizing an off-policy Q-network for credit backpropagation at the action-level (referred to as utterance-level in the original text, unified as action-level here for clarity). This off-policy Q-network enhances sample efficiency, especially when pre-collected data is available. At the token-level, ArCHer directly backpropagates action-level credits to each intra-action token using REINFORCE. Notably, in practice, ArCHer introduces a V-network at the action-level to facilitate computing action-level advantage values, using these advantage values (rather than Q-values) as terminal rewards at token-level, which may introduce additional inconsistency. Moreover, the use of multiple value estimation networks (including an optional token-level baseline value model besides the Q and V network) increases manual tuning efforts and may lead to cumulative bias and variance, potentially impacting stability.

We have made every effort to test ArCHer's performance in the VirtualHome environment, yielding preliminary experimental results as shown in Figure 2. In complex Entertainment tasks, ArCHer is second only to POAD, aligning with our theoretical expectations. However, in tasks such as Food Preparation where the methods’ performance gap was less pronounced, ArCHer performed poorly, likely due to instability in its system.

Furthermore, we applied insights from our theoretical analysis to enhance ArCHer's performance, i.e. removing the intra-action gamma, as depicted in ArCHer-BAD in Figure 2 (right one), showing improved results that validate the effectiveness of our insights. Nevertheless, due to the inherent challenges of excessive network complexity and difficult hyper-parameter tuning, ArCHer still falls short of matching POAD's performance.

### Figure 3: Case Study
Thanks for the insightful suggestions from reviewers 671a and 9Wzo, **we have conducted a case study for in-depth analysis to validate the effectiveness of BAD in terms of token-level credit assignment.**

**Case study settings.** We selected the last three states from the Food Preparation task, which form a complete subtask: "Heat the Pancake with Microwave". This serves as a simple yet practical case for our analysis. The corresponding transitions are as follows:

$$\mathcal{T}(s_0,\text{``open the microwave"})\rightarrow s_1$$

$$\mathcal{T}(s_0,\text{``close the microwave"})\rightarrow s_0$$

$$\mathcal{T}(s_1,\text{``put the pancake into the microwave"})\rightarrow s_2$$

$$\mathcal{T}(s_2,\text{``open the microwave"})\rightarrow s_2$$

$$\mathcal{T}(s_2,\text{``close the microwave"})\rightarrow \text{success with reward 1.0}$$

According to these transitions, the optimal trajectory to complete this subtask is (open the microwave, put the pancake into the microwave, close the microwave). Additionally, the maximum step length for the task is 5; if the task is not completed within 5 steps, it is considered a failure, resulting in a reward signal of -1.0.

Based on this, we first sampled 1,000 examples using a random policy. We then trained the critic to convergence using three different backup methods: TWOSOME (action-level Bellman backup), NTPO (naive token-level Bellman backup), and POAD (BAD). In Figure 3, we recorded the credit assignment results for each token in positive and negative actions under the First State and Last State conditions for the critics trained with these three methods.

The first two subfigures in Figure 3 show the credit assigned to each token in a symmetric (positive, negative) action pair by the BAD-learned critic in two different states, results confirm that the BAD critic can effectively assign most of the credit to the key tokens while rarely influencing other irrelevant tokens. The right one illustrates the volume of credits assigned to key tokens by POAD and NTPO, compared with the credit assigned to the entire action by TWOSOME, showing that POAD enjoys a far smaller discrepancy than NTPO.

### Figure 4 (a): Full-scale Fine-Tuning
We greatly appreciate the reviewer H4uQ for the valuable suggestion, we conducted an ablation comparing our method with the action-level baseline, TWOSOME, on both full-scale and LoRA fine-tuning settings, with results shown in Figure 4 (a).

### Figure 4 (b)
Thanks for the suggestion from JxkT. we are working hard to improve the readability of the Figure 2 in our original paper. Figure 4 (b) here is a current version of this diagram.

---
[1] Zhou, Yifei, et al. "Archer: Training language model agents via hierarchical multi-turn rl." arXiv preprint arXiv:2402.19446 (2024).

---

### Decision · Program_Chairs · 2024-09-25

**Decision:**

Accept (poster)

**Comment:**

This paper investigates reinforcement learning settings where sampling actions corresponds to sampling *sequences* of tokens from a suitably conditioned large language model. To address this challenge, the authors designed a method known as Policy Optimization with Action Decomposition (POAD) that allows agents' policies to be optimized at the token level rather than the action level. The paper introduces a theoretical framework called Bellman backup with Action Decomposition (BAD). BAD was explicitly designed to tackle discrepancies between action-level and token-level optimization. This is a key and necessary feature in this setting since actions are noew described as *sequences* of tokens, which introduces issues relating to credit assignment between all tokens linked to a particular action.

The reviewers raised a few concerns. Reviewer JxkT brought up issues with the notation and formalization adopted by the authors. To the best of my knowledge, the authors acknowledged these issues and addressed them appropriately. In response to these updates and improvements, JxkT thanked the authors for their thorough response and for addressing the points they brought up. JxkT also appreciated that the authors extended their empirical analyses to account for concerns mentioned by other reviewers—specifically, they included comparisons with additional baseline methods and provided further and detailed experiments. This latter limitation was also the main concern mentioned by H4uQ. Based on the rebuttal and the post-rebuttal discussion, this concern seems to have been adequately addressed from H4uQ's perspective.

Furthermore, reviewers 671a and 9Wzo recommended that POAD be compared with a recently proposed relevant method (ArCHer), which the authors did. All results were included in a pdf file attached to the rebuttal. The reviewers were satisfied with the new experiments (671a: "*(...) many thanks to the authors for thoroughly supplementing the key experiments and discussions*"; 9Wzo: "*Thanks for the detailed response. I think the authors have addressed most of my concerns, including empirical comparison with ArCHer, societal impact, and whether Fig. 2 is true*").

Although all reviewers brought up important concerns, and the consensus seemed to be that this paper is not necessarily strongly novel, it does provide the community with interesting insights on how to (at least partially) tackle the issue of credit assignment in settings where RL agents sample actions from an LLM. Based on the feedback provided by the reviewers, I do not see any critical flaws, missing comparisons or analyses, or methodological issues with this paper. More importantly, the extensive and fruitful discussion between the authors and the reviewers, as well as the reviewers' acknowledgment of the authors' contributions and findings, makes me confident that the NeurIPS community will benefit from the insights presented in this work.